# Enhancing Training Data Attribution with Representational Optimization

**Weiwei Sun**[1]    **Haokun Liu**[2]    **Nikhil Kandpal**[2]    **Colin Raffel**[2]    **Yiming Yang**[1]

[1]Carnegie Mellon University    [2]University of Toronto & Vector Institute

{sunnweiwei,haokunliu412,nkandpa2,craffel}@gmail.com

## Abstract

Training data attribution (TDA) methods aim to measure how training data impacts a model's predictions. While gradient-based attribution methods, such as influence functions, offer theoretical grounding, their computational costs make them impractical for large-scale applications. Representation-based approaches are far more scalable, but typically rely on heuristic embeddings that are not optimized for attribution, limiting their fidelity. To address these challenges, we propose AirRep, a scalable, representation-based approach that closes this gap by learning task-specific and model-aligned representations optimized explicitly for TDA. AirRep introduces two key innovations: a trainable encoder tuned for attribution quality, and an attention-based pooling mechanism that enables accurate estimation of group-wise influence. We train AirRep using a ranking objective over automatically constructed training subsets labeled by their empirical effect on target predictions. Experiments on instruction-tuned LLMs demonstrate that AirRep achieves performance on par with state-of-the-art gradient-based approaches while being nearly two orders of magnitude more efficient at inference time. Further analysis highlights its robustness and generalization across tasks and models. Our code is available at https://github.com/sunnweiwei/AirRep.

## 1 Introduction

The remarkable success of large language models (LLMs) has been demonstrated across a wide range of tasks. However, a fundamental question remains open in machine learning: *how does the behavior of LLMs depend on their training data?* More specifically, what training examples cause models to generalize well—or underperform—on specific inputs or tasks? Training Data Attribution (TDA), the process of measuring the impact of specific parts of the training data on the predictions of machine learning models, is an important step to answering this question [1]. TDA is crucial for ensuring transparency and accountability in AI systems by revealing how data influences model outputs.

Existing approaches to TDA can broadly be categorized as either *gradient-based* or *representation-based*. Gradient-based approaches, rooted in influence estimation [2, 3], aim to quantify the impact of individual training points on model predictions. Their core idea is to use gradients and the inverse Hessian of the loss function to make a first-order approximation of how a model's predictions would change if a certain example was removed from the training set [3]. While these methods are theoretically well-motivated, they are typically computationally expensive and rely on assumptions of loss convexity and model optimality — both of which are violated in modern neural networks [4, 5].

In contrast, representation-based methods estimate influence based on similarity in the representation space. The core assumption is that training examples whose representations are close to a test input are likely to have influenced its prediction [6]. Recent works have explored various notions of similarity based on model hidden states [7, 6], n-gram features [8], and text embeddings [9, 10]. Compared to gradient-based TDA, representation-based approaches are more computationally efficient and

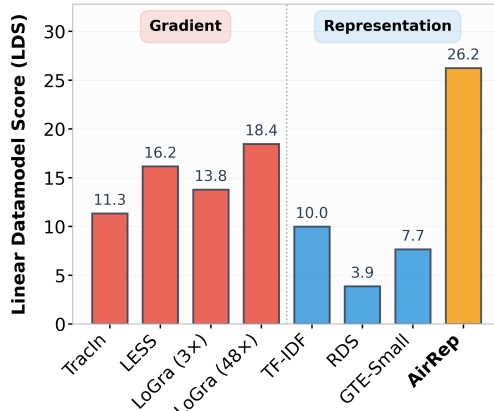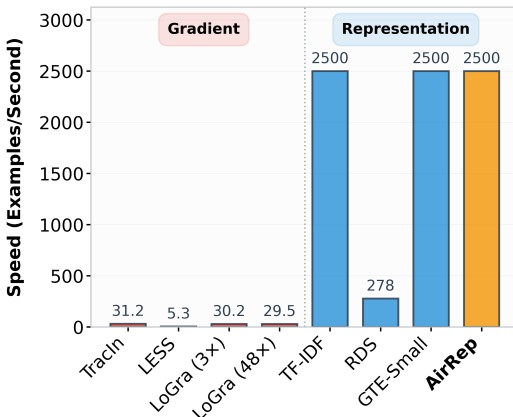

Figure 1: Performance comparison of gradient-based and representation-based training data attribution (TDA) approaches. **Left:** Average linear data model score (LDS) [1] on 4 unseen datasets (FLAN, Alpaca, Tulu, SafeRLHF). AirRep outperforms state-of-the-art gradient-based methods such as LoGra (which uses 48× more storage). **Right:** Inference speed (encoded examples per second on a single GPU). AirRep is nearly two orders of magnitude more efficient than gradient-based methods during inference.

scalable, making them well-suited for large-scale applications, such as curating pre-training data for LLMs [8, 11]. However, the quality of a representation-based method depends heavily on the selected feature space. This problem is compounded by the fact that existing work utilizes heuristically designed representations that are not tailored to the particular target task or model [12].

Finally, a persistent challenge in both method families is is how these methods are extended to estimate the collective influence of a group of training examples. Most past work extends single-example attribution methods to the group setting by simply summing over individual attribution scores within the group. However, this additive assumption fails to capture groupwise interactions and may lead to inaccurate influence estimation [13].

The advantages and shortcomings of existing methods raise a natural question: *Can we develop a method that provides the best of both worlds?*

We answer this question by introducing the Attentive Influence Ranking Representation (AirRep), a new representation-based approach that incorporates two novel mechanisms to improve performance over prior representation-based methods. First it uses a trainable encoder that produces task- and model-specific text representations, ensuring that the feature space is optimized for data attribution estimation. Second, it introduces an attention-based pooling mechanism that effectively aggregates multiple examples into a single representation for more accurate group influence estimation. AirRep is optimized to produce scores that accurately reflect the influence of particular groups of training examples on a model's predictions. To achieve this, we device a automatic pipeline to construct pairwise comparison dataset where data subsets are ranked based on their contributions to a model's predictions on specific target examples. We optimize AirRep on the dataset with a weighted pairwise ranking objective so that the learned representations and aggregation accurately represent the influence of data on the model's predictions.

Building on the setup of *datamodels* [1], we apply our approach on FLAN [14], an instruction-tuning dataset, and UltraChat [15], a large-scale dialogue generation dataset, and evaluate the model on five unseen instruction-tuning test set (FLAN, Alapca, Tulu, SafeRLHF) that do not appear in the training data. Figure 1 shows the main evaluation results. Notably, in the standard Linear Datamodeling Score (LDS) evaluation [1], AirRep significantly outperforms existing gradient-based approach [16], despite being ∼80× more computationally efficient and ∼50× more storage efficient at inference time. Our evaluation additionally shows consistent patterns across various downstream TDA tasks, including data selection, data source identification, and task classification. AirRep also exhibits strong generalizabilty to new tasks and models, which further underscoring the effectiveness of AirRep (Section 5). Finally, we show that the inference efficiency of AirRep can effectively amortizes the cost of AirRep training [17] (Section 9).

## 2 Preliminaries

We consider a framework that is applicable to different applications of TDA but focuses on the training of language models (LMs) as a concrete example. Let $S = (z_1, z_2, \ldots, z_n)$ represent a training set of $n$ data points, where $z_i$ is a training example that includes both an input and an output. In the context of language models, we refer to the input as a "prompt" and the output as a response. Furthermore, let $x$ denote a test example. The goal of TDA is to understand how the training examples in $S$ contribute to the model's prediction on $x$.

Formally, let $\theta$ represent the parameters of the LM. The model is fine-tuned on $S$, aiming to find

$$\theta^* = \arg\min_{\theta} \sum_{z_i \in S} \ell(z_i; \theta), \tag{1}$$

where $\ell(\cdot)$ denotes the cross-entropy token prediction loss. As language model training is non-convex, we use $\theta^*$ to denote the optimized parameters after fine-tuning, regardless of whether a true minimum has been found. We define the model's "*prediction outcome*" on $x$ as the cross-entropy loss computed on $x$, given by $r(x, S) = \ell(x; \theta^*)$, i.e. the evaluation loss on the test example $x$ for the model fine-tuned on the dataset $S$. Here, we use the cross-entropy loss instead of task-specific metrics like the accuracy, F1 score, or ROUGE [18] because the loss is a generic metric that is defined for all NLP tasks and has been found to be correlated with task-specific metrics [19].

A data attribution model, denoted by $f(x, S)$, aims to estimate the actual retraining outcome $r(x, S)$—i.e., the loss of the model on example $x$ after training on dataset $S$. Existing methods can be generally categorized into two groups: *gradient-based methods* and *representation-based methods*.

**Gradient-based methods**    Gradient-based methods largely stem from influence functions [2, 3], which approximate how $\theta$ would change in response to infinitesimal perturbations in the weighting of training instances. Specifically, influence functions quantify the influence from a single training example $z_i$ to an evaluation example $x$ with a closed form expression:

$$f_{\text{IF}}(x, z_i) = -\nabla_{\theta}\ell(x; \theta)^{\top}\, \mathbf{H}^{-1}\, \nabla_{\theta}\ell(z_i; \theta). \tag{2}$$

where $\mathbf{H}^{-1}$ is the inverse Hessian of the training loss wrt model parameters. The derivation of Eq. 2 is given by first-order Taylor approximation of $\ell(x; \theta)$ around $\theta^*$ [20] (see Appendix A for details).

The size of model neural networks makes the computation and inversion of the Hessian intractable. Consequently, practical influence function-based methods often use an approximation of the Hessian [12, 21]. Recent techniques have also considered techniques like gradient projection [16] and model ensembling [22, 23]. As a representative approach in LLM application, LoGra [16] define the influence of a training example $z_i$ on $x$ is computed as:

$$f_{\text{GD}}(x, z_i) = \phi(x)^{\top} \cdot \phi(z_i), \tag{3}$$

where $\phi(z)$ is the projected, Hessian-corrected, and unit-normalized gradient for the input example $z$, given model parameters $\theta$:

$$\phi(z) = \text{norm}\big[\mathbf{H}_{\hat{\theta}}^{-\frac{1}{2}} \nabla_{\hat{\theta}}\ell(z; \theta)\big]_2, \tag{4}$$

where $\text{norm}\big[\mathbf{z}\big]_2 = \frac{\mathbf{z}}{\|\mathbf{z}\|_2}$ is a unit normalization operation [24]; $\hat{\theta}$ denotes the trainable weights of a LoGra module [16] with PCA initialization for gradient projection; $\nabla_{\hat{\theta}}\ell(z; \theta)$ is the gradient of the loss with respect to $\hat{\theta}$; and $\mathbf{H}_{\hat{\theta}}^{-\frac{1}{2}}$ is computed on $\hat{\theta}$ and is approximated based on the Kronecker-Factored Approximate Curvature (KFAC) algorithm [21]. Further, under first-order approximation, the group influence $f_{\text{GD}}(x, S)$ is estimated as the summation of individual influences.

**Representation-based methods**    Representation-based methods encode examples as embedding vectors, eliminating the need for gradient computation. The influence score of a training example $z_i$ to $x$ can then be computed as:

$$f_{\text{Rep}}(x, z_i) = \text{Enc}(x)^{\top} \cdot \text{Enc}(z_i), \tag{5}$$

where $\text{Enc}(z)$ denotes a continuous embedding of input $z$ encoded by encoder model $\text{Enc}$. Group influence is usually defined as summation of individual influence under a linear assumption [1].

In term of the encoding function $\text{Enc}$, existing work in the text domain has investigated different heuristic designs including (last-layer) hidden states of a language model [7, 6], n-gram representations [8], TF-IDF representations [25, 26], and text embeddings [11, 27]. See Appendix C for further comparison of the two types of TDA methods.

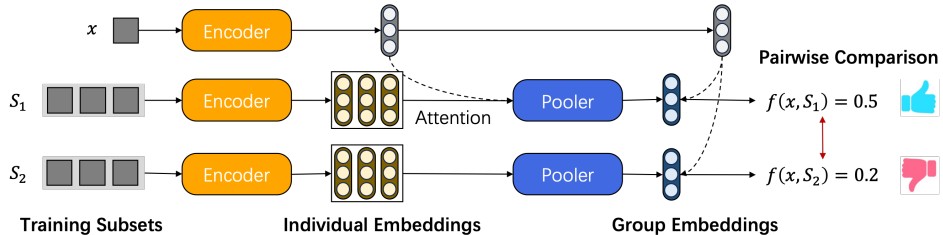

Figure 2: Model Architecture and Optimization. The test example $x$ and the training subsets $S_1$ and $S_2$ are encoded by an encoder with a pooler to obtain embeddings. The score is computed as the inner product of the embeddings. The overall model is trained based on pairwise comparisons to distinguish the usefulness of different subsets with respect to the test example $x$.

## 3 AirRep

To bridge the gap between gradient- and representation-based data attribution, we introduce AirRep, a novel data-driven approach to enhance representational influence models. At a high level, AirRep consists of a trainable encoder $\mathrm{Enc}$ for embedding input examples and a pooling layer $\mathrm{Agg}$ that aggregates the representation of individual examples to produce a group representation. The influence score between a training set $S$ and a target example $x$ is then computed as

$$f_{\mathrm{AirRep}}(x, S) = \mathrm{Enc}(x)^\top \cdot \mathrm{Agg}(\mathrm{Enc}(z_i) \mid z_i \in S) \tag{6}$$

Specifically, AirRep incorporates two novel mechanisms to improve performance over standard representation-based methods: First, we introduce an effective attention-based pooling mechanism that better reflects group influence effect (Section 3.1). Second, we optimize the model on auto-generated data so that AirRep's score better reflects the underlying models predictions (Section 3.2).

### 3.1 Attention-based Pooling

Recall that data attribution methods typically include a mechanism for aggregating the contributions of elements of $z_i \in S$ when performing group influence. For this purpose, AirRep uses a pooling layer to aggregate the representations $\mathrm{Enc}(z_i)$ into a single embedding. The design of this pooling layer is crucial for modeling the relationships between group of data samples.

In AirRep, we propose a simple yet effective pooling techniques: attention-based influence pooling. Specifically, AirRep's aggregation method incorporates an attention-like operation to capture interactions between elements of $x$ and $S$:

$$f_{\mathrm{AirRep}}(x, S) = \mathrm{Enc}(x)^\top \cdot \sum_{i=1}^{n} \alpha_i \, \mathrm{Enc}(z_i),$$
$$\text{where} \quad \alpha_i = \frac{\exp(|\mathrm{Enc}(x)^\top \cdot \mathrm{Enc}(z_i)|)}{\sum_{j \in [n]} \exp(|\mathrm{Enc}(x)^\top \cdot \mathrm{Enc}(z_i)|)}. \tag{7}$$

where $\mathrm{Enc}(x)$ denotes the sentence embedding predicted by a BERT-based sentence encoder [28, 29], $\alpha_i$ denotes the attention score of example $z_i$, and $|\cdot|$ denotes the absolute value operation.

**Remark** Empirically, we observe that influence scores are sparse, where each test example relies on only a few training points, while others add noise. This supports the need for selective pooling, consistent with prior findings [1, 12, 30]. Additionally, the proposed attention-based influence pooling can also related to high-order group influence functions. Specifically, Basu et al. [13] showed that second-order terms capture additional relationship between samples. Extending this, we show that high-order group influence introduces sample-wise weights, akin to our attention-based pooling (see Appendix B.1 for more discussion).

### 3.2 Optimization

Both the $\mathrm{Enc}$ and $\mathrm{Agg}$ components of AirRep are trainable modules that together take as input $x$ and $S$ and output a scalar score $f(x, S)$. Ultimately, the objective of data attribution is that $f(x, S)$

is correlated to $r(x, S)$, i.e., the actual prediction on $x$ of model trained on $S$. Since AirRep is overall trainable, we therefore formulate a training objective that aims to make $f(x, S)$ reflect $r(x, S)$. Given a ground-truth collection of $r(x, S)$ scores for different datasets $S$, we could in principle train $f(x, S)$ to match these scores as a regression problem. However, in data attribution we care less about matching the exact values of $r(x, S)$ than we do reflecting the *ranking* of different example groups (datasets), since ultimately we aim to answer questions like "*which group had the largest influence on this example?*". We therefore formulate training as a pairwise ranking problem – that is, given two different data subsets $S_1$ and $S_2$, whether the score of $f(x, S_1)$ and $f(x, S_2)$ reflect the true data preference $r(x, S_1)$ and $r(x, S_2)$. In this section, we will first introduce the data generation pipeline we used for collecting ground-truth $r(x, S)$ scores and then describe the specific training objective we formulated.

**Data Generation**   To construct data for training AirRep, we assume access to a large corpus of example datapoints. Then, we generate cross-validation-style data and compute ground-truth attribution scores. Specifically, we first sample $N_v$ examples as a validation split and $N_t$ examples as a training split. Then, from the training split, we randomly sample $M$ subsets of data with replacement, denoted as $\mathcal{S} = \{S_1, S_2, \ldots, S_M\}$, where each subset $S_i$ contains $n$ training examples. For each training subset $S_i$, we finetune a language model on it as in Eq. 1 to obtain a model checkpoint $\theta_i$. We then evaluate the trained model on each example $x$ in the validation subset to calculate the loss: $\ell(x; \theta_i)$. After training $M$ models and calculating the loss, we compute the negative normalized loss for each $x$ in the validation set as

$$\hat{r}(x, S_i) = -\frac{\ell(x; \theta_i) - \text{Mean}\left(\{\ell(x; \theta_j) \mid j \in [M]\}\right)}{\text{Var}\left(\{\ell(x; \theta_j) \mid j \in [M]\}\right)}, \tag{8}$$

where the original loss $\ell(x; \theta_i)$ is normalized by the mean and variance across models to stabilize the training signal, ensuring that the distribution of scores for each target $x$ is on a similar scale.

**Training Objective**   Given a target example $x$ from our validation set, $M$ training subsets $\mathcal{S} = \{S_1, S_2, \ldots, S_M\}$, and the corresponding normalized loss values $\{\hat{r}(x, S_i); i \in [M]\}$, AirRep estimates the normalized loss values for each subset as $f(x, S_i)$. For brevity, we use $r_i$ and $f_i$ to represent these scores, respectively. Our proposed training objective is a pairwise ranking loss that encourages the difference between the model-predicted scores, $f_i - f_j$, to better align with the difference in precomputed scores, $r_i - r_j$.

In practice, the label $r_i$ is noisy due to the stochastic nature of LM training. To mitigate the impact of noisy labels, we adopt a weighted pairwise ranking loss, inspired by importance reweighting [31, 32], which assigns lower weights to uncertain labels while prioritizing more reliable ones:

$$\mathcal{L}(x, \mathcal{S}) = -\sum_{i,j \in M} \mathbb{1}_{r_i > r_j} \, w_{i,j} \, \log \sigma(f_i - f_j),$$

$$\text{where} \quad w_{i,j} = \begin{cases} 0, & \text{if } |r_i - r_j| < T_{\min}, \\ \min\{|r_i - r_j|, T_{\max}\}, & \text{if } T_{\min} \leq |r_i - r_j|. \end{cases} \tag{9}$$

$\sigma$ is sigmoid function, $w_{i,j}$ is a weighting function that depends on the absolute difference in ground truth scores, $|r_i - r_j|$, and is clipped using thresholds $T_{\min}$ and $T_{\max}$ to avoid training on $i, j$ pairs with incorrect ordering and to mitigate the impact of outliers.

## 4   Experimental Setup

**Model**   Our experiments focus on LM finetuning, using the Qwen2.5 model family [36] as our base LMs. During training, we start with the base LM and fine-tune it using a batch size of 32 and the AdamW optimizer [37] with a learning rate of 2e-5 for two epochs.

**Data Generation**   For AirRep training, we utilize two datasets, each representing distinct scenarios: instruction tuning on standard NLP tasks and training to improve conversational abilities: (i) FLAN [14], an instruction-tuning dataset for language models, and (ii) UltraChat [15], a large-scale dataset of instructional conversations covering a wide range of topics. To generate training signal, we set $N_v = 10^4$ and $N_t = 10^5$. The training subsets number is $M = 100$, with each subsets containing $n = 1,000$ samples. We construct 100 cross-validation instances. Thus, in total, the data includes

Table 1: LDS Evaluation Results of Qwen2.5-0.5B on four Datasets. *Avg* denotes the average score. *Dim* refers to the dimensionality of the embeddings (which corresponds to storage size).

| Method | Dim | Avg | FLAN | Alpaca | Tulu | SafeRLHF |
|---|---|---|---|---|---|---|
| TracIn [22] | 18432 (48×) | 11.33 | 14.75 | 9.21 | 10.75 | 10.60 |
| LESS [25] | 8196 (21×) | 16.16 | 16.40 | 9.59 | 13.02 | 25.63 |
| LoGra [16] | 1152 (3×) | 13.78 | 13.32 | 6.87 | 10.16 | 24.76 |
| LoGra [16] | 18432 (48×) | 18.45 | 19.75 | 12.38 | 14.88 | 26.82 |
| Dsdm [33] | 18432 (48×) | 18.02 | 19.67 | 12.15 | 14.31 | 25.94 |
| TF-IDF [34] | - | 9.98 | 2.52 | 7.24 | 5.24 | 24.94 |
| DSIR [8] | - | -0.02 | 0.49 | 2.01 | -0.49 | -2.10 |
| RDS [6] | 896 (2.3×) | 3.86 | 0.74 | 0.87 | 1.89 | 11.94 |
| GTE-Small [35] | 384 (1×) | 7.65 | 0.92 | 1.74 | 1.14 | 26.80 |
| **AirRep (Ours)** | 384 (1×) | **26.23** | **21.11** | **22.58** | **15.14** | **46.08** |

$10^4$ unique training subsets and $10^7$ training examples. The Qwen2.5-0.5B LM is then fine-tuned on these training subsets and evaluated on the corresponding validation subset to obtain the label $r(\cdot, \cdot)$.

**AirRep Training Details**    We initialize AirRep using the GTE-Small, a 30M parameter embedding model [35], and apply a randomly initialized projection matrix on top. AirRep is trained separately on FLAN and UltraChat. To construct the data for each training step of AirRep, we randomly select one cross-validation instance, then sample 1,000 examples from its validation subset and 32 training subsets. Distributed training is employed to maximize GPU memory utilization. The clipping thresholds, $T_{\min}$ and $T_{\max}$, are set to 0.1 and 5.0, respectively. The model is optimized for up to 2,000 steps using the AdamW optimizer with a learning rate of $1 \times 10^{-4}$.

**Evaluation Datasets**    We use the following datasets for evaluation: (i) *FLAN* [14]: The evaluation data of FLAN contains 66 NLP tasks spanning diverse categories. (ii) *Alpaca* [38]: An instruction-tuning dataset generated by OpenAI's text-davinci-003 model. (iii) *Tulu* [39]: An instruction-tuning dataset comprising diverse data sources. (iv) *SafeRLHF* [40]: A dataset for safety alignment of large language models, where each response is labeled as either safe or unsafe. Note that each evaluation data contain a test set and a training set. We ensure that all evaluation data remain excluded from AirRep's optimization data to guarantee that the evaluation results reflect AirRep's generalization to unseen data. Appendix D provides additional dataset details.

**Baselines**    We compare AirRep with representative gradient-based and representation-based methods. The gradient-based baselines we consider are as follows: (i) LoGra [16], an optimized version of the influence function as described in Eq.4, based on its implementation in the Logix software library. (ii) TracIn [22], which leverages the dot product of gradient vectors. (iii) LESS [25], which computes projected gradients on LoRA weights and adjusts gradients using the AdamW states. (iv) Dsdm [33], an efficient implementation of Trak [12] for LLM tasks.

The representation-based baselines we consider are as follows: (i) TF-IDF [34], (ii) DSIR [8], hashed n-gram features, (iii) RDS [6], which utilizes the last-layer hidden states of LMs, and (iv) GTE [35], a text embedding model trained on text similarity and retrieval tasks which was also used as the base pre-trained model for AirRep. See Appendix E for more implementation details.

## 5 Evaluation of Data Attribution

**Setup**    To evaluate these data attribution methods, we follow [1] and use the linear datamodeling score (LDS) as our evaluation metric. For each evaluation dataset, we sample 100 different random subsets $(S_1, \cdots, S_{100})$ of its training set, each containing $n$ training examples, and train the target LM on each of these subsets. For each example $x$ in the test set, we approximate the expectation of the model prediction $r(x, S_i)$ as the evaluation cross-entropy loss. Given the score $f(x, S_i)$ predicted by the data attribution model, the LDS is computed as the Spearman rank correlation between the true and estimated influence scores: $\rho\big(\{r(x, S_i) : i \in [100]\}, \{f(x, S_i) : i \in [100]\}\big)$. Finally, the LDS on the dataset is averaged across all test examples.

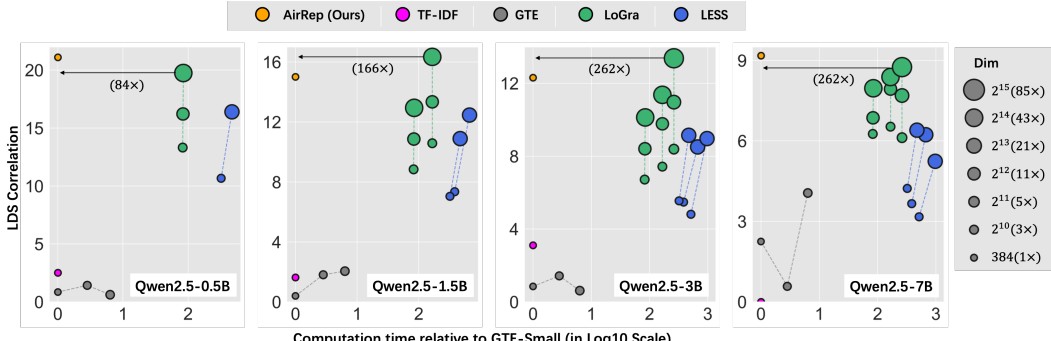

Figure 3: LDS correlation scores on FLAN vs. inference-time cost and storage for various TDA methods across Qwen2.5 models (0.5B–7B). Computation time (Log10 scale) is measured relative to GTE-Small on the same machine; marker size reflects storage (smaller = more efficient). Each method has multiple points for different model/dimension settings.

**Results** Table 1 presents the LDS evaluation results for attributions computed on the Qwen2.5-0.5B model's predictions across four evaluation datasets. AirRep achieves the best performance across all evaluation datasets. Notably, it outperforms the competitive gradient-based approach LoGra [16], despite being $80\times$ more computationally efficient and $50\times$ more storage efficient. Additionally, previous representation-based methods exhibit low correlation scores in this evaluation, underscoring the importance of task-aware optimization of representations.

Figure 3 illustrates the LDS correlation when applying TDA methods to different target language models, ranging from Qwen2.5-0.5B to Qwen2.5-7B. For gradient-based methods such as LoGra and LESS, the proxy model can be adjusted—i.e., a smaller LM can be used as a proxy to compute influence scores for larger target LMs [25]. Additionally, the projection dimensionality can be modified to balance computation, storage, and performance. In all cases, we use the same AirRep model trained exclusively on data generated by Qwen2.5-0.5B. Notably, AirRep demonstrates strong performance when applied to larger target LMs. Specifically, it consistently achieves results comparable to the best gradient-based methods, which require significantly more resources (e.g., $84\times$ to $262\times$ more computation time and $50\times$ to $80\times$ more storage). These results highlight AirRep's potential for efficient and effective data attribution. AirRep also demonstrates strong generalization across different data sizes and target LM types (Figure 8 and Figure 9).

## 6 Evaluation of Data Selection

**Setup** One use of data attribution is data selection, where we pick out the highest value training data and train only on these samples. To evaluate each attribution method's utility in this setting, we use the FLAN collection of datasets. For each task in FLAN, we use each TDA method to select a high-value subset using a greedy selection strategy. First, given the scores predicted by TDA models, we rank all training examples based on their highest score with respect to the test set samples and retain the top 1,000 examples. We then train LMs on the selected subset and evaluate their performance on the test set. The evaluation metric is the F1 score between the LM-generated outputs and the ground-truth answers.

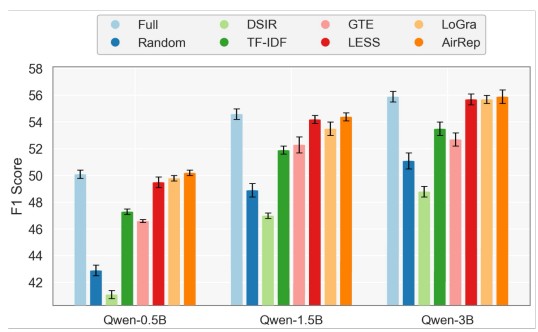

Figure 4: Evaluation Results of Data Selection. We report the average F1 score of 66 tasks in FLAN, obtained by training Qwen2.5 LM of different sizes on the top-1000 selected examples for each task.

**Results** Figure 4 presents the results of using six TDA methods for data selection, along with a random data selection baseline, evaluated in terms of F1 score and selection cost. Key findings

include: (i) Across all model scales, all methods (except DSIR) outperform random selection. (ii) Gradient-based approaches (LESS and LoGra) achieve better performance than representation-based methods (TF-IDF and GTE); however, they are computationally expensive—the time required to select a subset exceeds the time needed to train the model on the full data, (iii) AirRep performs comparably to gradient-based methods and full-data training, while significantly outperforming other representation-based methods in effectiveness, all while maintaining efficiency.

## 7 Evaluation of Data Classification

**Setup** TDA methods have been applied to measuring data similarity for the purpose of model explainability [6]. To evaluate the effectiveness of such explanations, Hanawa et al. [6] introduced the "Identical Class Test" for image classification, which assesses whether the most influential training example identified by TDA methods belongs to the same class as the test instance.

We extend this test to LM fine-tuning and formulate the "data classification" task, which measures the accuracy of TDA methods by evaluating whether the top-retrieved training examples share the same label as the test example. We consider the following sources of labels for classification: (i) FLAN and Tulu: FLAN comprises 66 NLP tasks, while Tulu is a collection of 12 source datasets (e.g., ShareGPT, CoT, Code-Alpaca). We use the task type or source dataset as a label, expecting that training data from the same task or source as the test example will be more valuable. (ii) SafeRLHF: SafeRLHF contains both safe and unsafe data annotated with safety labels. We use the safety label, expecting that harmful training data are more likely to lead to harmful generations at test time.

**Results** Table 2 presents the evaluation results of LoGra, GTE-Small, and AirRep.[1] The results indicate that AirRep significantly improves data classification accuracy on FLAN compared to GTE-Small (increasing from 50.59 to 86.41) and also outperforms LoGra. Notably, since AirRep is trained without using any data labels, this suggests that the model can learn to represent task-related

Table 2: Accuracy of Data Classification.

| Method | FLAN | Tulu | SafeRLHF |
|---|---|---|---|
| LoGra (Dim 1152) | 71.61 | 76.60 | 78.00 |
| LoGra (Dim 18432) | 85.44 | 86.00 | 83.20 |
| GTE-Small | 50.59 | 76.60 | **90.60** |
| AirRep | **86.41** | **88.20** | 87.20 |

information in an unsupervised manner. A similar trend is observed for Tulu, where AirRep demonstrates superior performance in identifying the data source. For SafeRLHF, AirRep outperforms LoGra but lags behind GTE-Small, likely because AirRep's training data, generated from UltraChat, does not contain harmful content and thus lacks the necessary learning signal.

## 8 Ablation Study

We conduct an ablation study to evaluate each component's contribution. Figure 5 presents the results as the average LDS score across the four datasets we consider. Starting with GTE (Small), which gets a score of 7.65, we first optimize the encoder while keeping mean pooling instead of adding the attention pooling layer. This results in AirRep w/o Attention, which achieves a score of 19.82—an improvement of 12.17—demonstrating the effectiveness of encoder optimization. Next, when we introduce the attention pooling layer and jointly optimize the model,

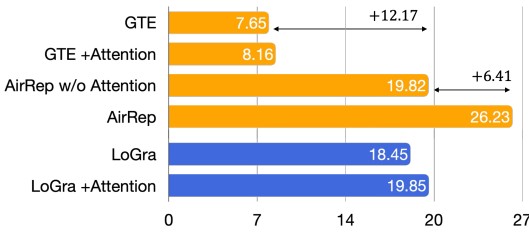

Figure 5: Ablation Study results: average LDS score on FLAN, Alpaca, Tulu, and SafeRLHF.

we observe a further improvement of 6.41. Additionally, we examine whether simply adding a softmax attention pooling to baseline to re-weight individual influences can enhance performance. From the results of GTE + Attention and LoGra + Attention, we observe only marginal improvements. This suggests that while weighting data is beneficial, optimizing the weighting distribution is also necessary to achieve significant gains. Finally, we observe that adding attention pooling not only improves group influence estimation but also leads to a better underlying representations.

---

[1]Here we focus on a subset of most competitive or directly relevant baselines, see Appendix H.2 for more.

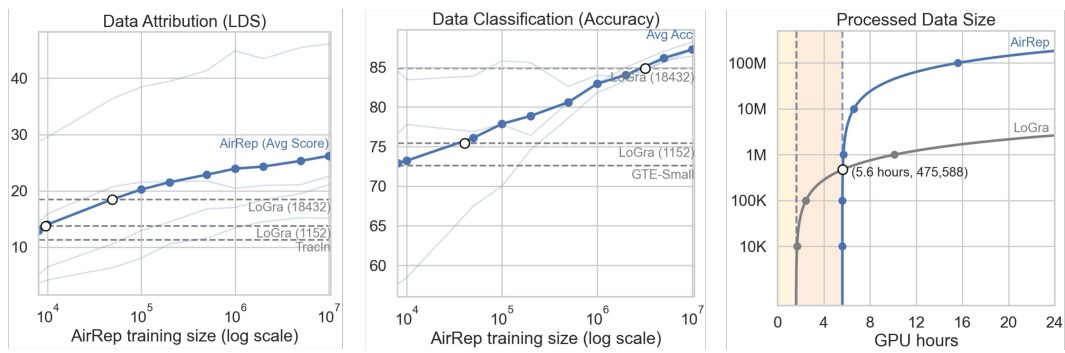

Figure 6: **Left:** AirRep training data size vs. LDS. **Middle:** AirRep training data size vs. Data Classification Accuracy. **Right:** GPU hours vs. number of data processed. AirRep incurs higher training costs due to data generation (yellow region) and model training (orange region), but is more inference-efficient.

## 9 Amortizing Training Cost

Compared with gradient-based methods, AirRep is more efficient at inference but requires training on auto-generated data, which incurs additional cost. Our evaluation above shows that FLAN- or UltraChat-pretrained AirRep generalizes well on new TDA data and tasks. We now consider another setting where AirRep is re-trained from scratch for each TDA task, and analyze how its inference efficiency can amortize its training cost [17].

Figure 6 (left and middle) shows AirRep's average attribution and classification scores with different training data sizes (measured by number of datapoints used in training). Light curves indicate per-test-set scores. As the training size increases, AirRep achieves better performance. In particular, AirRep outperforms LoGra (1152)—which has a comparable embedding size—when trained with around $10^5$ examples. We refer to this AirRep model as the "*crossover checkpoint*".

Then, Figure 6 (right) compares the number of training examples that can be processed under different A100 GPU hour budgets by AirRep (crossover checkpoint) and LoGra, accounting for both data generation and model training costs. LoGra completes model training earlier, but AirRep—despite longer training time—achieves nearly two orders of magnitude more throughput once model is trained. Notably, a turning point appears around 475K examples, where both methods take 5.6 GPU hours. Beyond this, even with retraining, AirRep is faster than LoGra. For example, with 24 GPU hours, AirRep can attribute over 100M examples, whereas LoGra can process only around 2M. This highlights the advantage of AirRep for large-scale TDA scenarios. We additionally emphasize that, as shown in Section 5, AirRep can be transferred from a small model to larger models and still attain stronger performance than all baselines, so the "re-training from scratch" results in this section represent worst-case results for AirRep.

**Amortizing Across Models** We evaluate Air-Rep, trained with Qwen-0.5B-generated supervision, using different target LMs (Qwen-0.5B, Qwen-7B, Llama-1B, Qwen3-0.6B, Qwen3-0.6B (Thinking), TinyLlama-1B, GPT-2) on FLAN. Results in Figure 7 show that across all targets, AirRep consistently surpasses Lo-Gra, even when LoGra computes gradients on the target LMs. We also observe that LoGra is sensitive to the proxy used for gradient calcula-tion—when the target model differs significantly (e.g., GPT-2), its performance drops notably, whereas AirRep maintains strong performance. This indicates the robust generalization of Air-Rep and that its training cost can be amortized across different target language models.

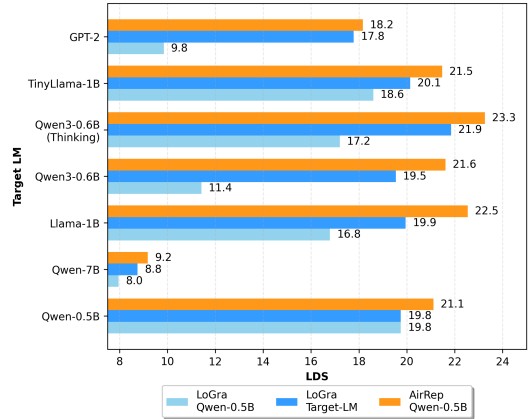

Figure 7: Evaluate with different target LMs. Ar-Rep shows strong generalization.

## 10 Related Work

Most existing work in TDA focuses on gradient-based methods, such as Influence Functions [3] and their subsequent optimizations, including group influence estimation [41, 13], Hessian approximation [42, 21, 43], gradient normalization [24, 25], gradient projection [22, 16], model ensembling [12, 33, 44], and distillation of the proxy influence models [45–47]. However, studies have found that the derivation of gradient-based methods may rely on incorrect assumptions in large neural networks [48, 4]. Additionally, gradient-based methods suffer from the computational expense of gradient calculations and Hessian inverses [21, 33].

Alternatively, representation-based approaches for TDA measure data influence in the representation space [7, 49, 6, 9], offering computational and storage efficiency. However, current methods rely on heuristically designed representations without task-aware and model-specific optimization, often leading to suboptimal results [12]. Some research on TDA explores simulation-based methods such as datamodels [50–52, 1], which are usually not tractable for LLM applications due to their computational expense.

TDA methods have been applied to various purposes in machine learning, such as identifying mislabeled training data points [3, 53], detecting data poisoning attacks [54, 55], and guiding data augmentation [56, 57]. TDA methods have also been used for understanding LLM generalization [21] and learning processes [30, 58]. Another notable line of related work is data selection for LLMs, which focuses on curating data to optimize test performance [59]. TDA-like functions for measuring relationships between data have been widely adopted in data selection pipelines to enhance the quality and diversity of the selected data. Both gradient-based [60, 25, 47] and representation-based [8, 61, 62, 10] TDA approaches have been applied. Meanwhile, studies have explored using text embedding models to select in-context examples for LLM prompting [63–65].

## 11 Conclusion

This paper introduces a new approach to training data attribution called AirRep, which combines the advantages of task-driven optimization in gradient-based approaches with the efficiency of representation-based methods. AirRep represents training data as embeddings and scores their influence using inner-product similarity in the representation space. AirRep's novel attention-based pooling mechanism captures the group effects, and the entire model is optimized using a novel proposed weighted pairwise ranking objective over automatically generated influence signals. Evaluation on LLM fine-tuning demonstrates the effectiveness and generalization of AirRep.

Limitations of this work include the added cost of model training and evaluation limited to the LLM fine-tuning stage. While our current evaluation focuses on language tasks, the design of AirRep is modality-agnostic—it relies only on the availability of suitable encoders and the ability to estimate loss differences. In principle, this framework could be applied to other domains such as vision or multimodal settings. For future work, we plan to extend our method to additional scenarios, such as the pre-training stages of LLMs and multi-modal model training.

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

# A   Deriving the Influence Function

Let $\theta^*$ denote the minimizer of the empirical risk:

$$\theta^* = \arg\min_\theta L(\theta),$$

where $L(\theta) := \frac{1}{n} \sum_{i=1}^n \ell(z_i; \theta)$.

Assume that $\ell$ is twice differentiable and strongly convex in $\theta$. Thus, the Hessian

$$\mathbf{H} := \nabla_\theta^2 L(\theta) = \frac{1}{n} \sum_{i=1}^n \nabla_\theta^2 \ell(z_i; \theta)$$

exists and is positive definite, ensuring the existence of $\mathbf{H}^{-1}$.

The perturbed parameters $\theta_{\epsilon,z}^*$ are defined as:

$$\theta_{\epsilon,z}^* = \arg\min_\theta \left( L(\theta) + \epsilon \ell(z, \theta) \right).$$

Define the parameter change due to the perturbation as $\Delta_\epsilon = \theta_{\epsilon,z}^* - \theta^*$. Since $\theta^*$ does not depend on $\epsilon$, the quantity we seek can be written in terms of $\Delta_\epsilon$:

$$\frac{d\theta_{\epsilon,z}^*}{d\epsilon} = \frac{d\Delta_\epsilon}{d\epsilon}.$$

Because $\theta_{\epsilon,z}^*$ is a minimizer, its first-order optimality condition is:

$$0 = \nabla L(\theta_{\epsilon,z}^*) + \epsilon \nabla \ell(z, \theta_{\epsilon,z}^*).$$

For small $\epsilon$, using the Taylor expansion around $\theta^*$:

$$0 \approx \nabla L(\theta^*) + \epsilon \nabla \ell(z, \theta^*) + \left[ \nabla^2 L(\theta^*) + \epsilon \nabla^2 \ell(z, \theta^*) \right] \Delta_\epsilon.$$

Neglecting higher-order terms, this reduces to:

$$\Delta_\epsilon \approx - \left[ \nabla^2 L(\theta^*) + \epsilon \nabla^2 \ell(z, \theta^*) \right]^{-1} \left[ \nabla L(\theta^*) + \epsilon \nabla \ell(z, \theta^*) \right].$$

Since $\theta^*$ minimizes $L(\theta)$, it satisfies $\nabla L(\theta^*) = 0$. Dropping higher-order terms in $\epsilon$ yields:

$$\Delta_\epsilon \approx -\mathbf{H}^{-1} \nabla \ell(z, \theta^*) \epsilon.$$

Thus, the derivative of $\theta_{\epsilon,z}^*$ with respect to $\epsilon$ at $\epsilon = 0$ is:

$$\left. \frac{d\theta_{\epsilon,z}^*}{d\epsilon} \right|_{\epsilon=0} = -\mathbf{H}^{-1} \nabla \ell(z, \theta^*).$$

To compute the influence of upweighting $z$ on the loss at a test point $x$, we use the chain rule:

$$\left. \frac{d\ell(x, \theta_{\epsilon,z}^*)}{d\epsilon} \right|_{\epsilon=0} = \nabla_\theta \ell(x, \theta^*) \left. \frac{d\theta_{\epsilon,z}^*}{d\epsilon} \right|_{\epsilon=0}$$
$$= -\nabla_\theta \ell(x; \theta^*) \mathbf{H}^{-1} \nabla_\theta \ell(z; \theta^*).$$

The influence function is therefore given by:

$$f_{\text{IF}}(x, z) = -\nabla_\theta \ell(x; \theta^*) \mathbf{H}^{-1} \nabla_\theta \ell(z; \theta^*).$$

# B  Deriving the Group Influence Function

For group influence function, we interest in removing a group of training examples.

Let $\theta^*$ denote the minimizer of the empirical risk:

$$\theta^* = \arg\min_\theta L(\theta),$$

where $L(\theta) := \frac{1}{n}\sum_{i=1}^n \ell(z_i; \theta)$.

By up-weighting a subset of examples $S = (z_1, \cdots, z_m)$, we define the new objective:

$$L_S(\theta) = \frac{1}{n}\left(\sum_{z_i \notin S}(1 - \hat{\epsilon})\ell(z_i, \theta) + \sum_{z_i \in S}(1 + \epsilon)\ell(z_i, \theta)\right)$$

where $\hat{\epsilon} = \frac{m}{n-m}\epsilon$. We can see if $\epsilon = 0$ we get the original loss function $L(\theta)$ (where none of the training samples are removed) and if $\epsilon = -1$, we get the loss function where samples in $S$ are removed from training.

The perturbed parameters $\theta^*_{\epsilon,S}$ can be written as:

$$\theta^*_{\epsilon,S} = \arg\min_\theta L_S(\theta).$$

Define the parameter change as $\Delta_\epsilon = \theta^*_{\epsilon,S} - \theta^*$. Since $\theta^*$ does not depend on $\epsilon$, the quantity we seek to compute can be written in terms of $\Delta_\epsilon$:

$$\frac{d\theta^*_{\epsilon,S}}{d\epsilon} = \frac{d\Delta_\epsilon}{d\epsilon}.$$

## B.1  Group Influence Function

Extended from Basu et al. [13] results, we show that the structure of the high-order group influence function is given by:

**Theorem B.1** (Group Influence Function). *Let* $\phi(z) = H_{\boldsymbol{\theta}}^{-\frac{1}{2}}\nabla_\theta\ell(z; \theta)$, *and assume* $\nabla_\theta^k\ell(z; \theta)$ *for* $k \geq 3$ *is negligible. Then the* $k$-*order group influence function is given by:*

$$f_{IF}^{(k)}(x, S) = \phi(x)\sum_{z_i \in S}\sum_{t=1}^k c_t^{(k)}\alpha_t(z_i)\,\phi(z_i), \tag{10}$$

*where:* $c_t^{(k)}$ *are constants at the* $t$-*th order, dependent on the meta-information of the distribution of* $S$ *and* $\alpha_t(z)$ *is defined as:*

$$\alpha_t(z) = \begin{cases} 1, & \text{if } t = 1, \\ \phi(z)\sum_{z_j \in S}\alpha_{t-1}(z_j)\phi(z_j), & \text{if } t \geq 2. \end{cases}$$

From Eq. 10, we observe that the interaction of data samples exhibits an attention structure, with $\alpha_t(z_i)$ defining the weights of each data sample contributing to the final output. Intuitively, when $k > 1$, examples $z_i$ that are closer to the group effect are assigned higher weights. We find the theoretical analysis of high-order group influence derived a similar notion of "attention-based influence pooling" in AirRep, though AirRep's design can be viewed as a simplified version of B.1. These analysis also motivate us to explore more complicated influence pooling mechanism in our future work.

The following section provide the derivation of B.1.

## B.2  Deriving the First-Order Group Influence Function

Since $\theta^*_{\epsilon,S}$ minimizes $L_S(\theta)$, the first-order optimality condition is:

$$0 = \nabla_\theta L_S(\theta^*_{\epsilon,S}).$$

For small $\epsilon$, expand around $\theta = \theta^*$ and $\epsilon = 0$:

$$0 \approx \nabla_\theta L_S(\theta^*) + \left.\frac{\partial \nabla_\theta L_S(\theta^*)}{\partial \epsilon}\right|_{\epsilon=0} \epsilon + \nabla_\theta^2 L_S(\theta^*)\, \Delta_\epsilon.$$

At $\epsilon = 0$, the perturbed objective reduces to the original loss, so:

$$\nabla_\theta L_S(\theta^*)\big|_{\epsilon=0} = \nabla_\theta L(\theta^*) = 0.$$

The Hessian at $\epsilon = 0$ is:

$$\nabla_\theta^2 L_S(\theta^*)\big|_{\epsilon=0} = \nabla_\theta^2 L(\theta^*) \equiv \mathbf{H}.$$

The first-order change in the gradient with respect to $\epsilon$ at $\theta = \theta^*$ is:

$$\left.\frac{\partial \nabla_\theta L_S(\theta^*)}{\partial \epsilon}\right|_{\epsilon=0} = \frac{1}{n-m} \sum_{z_i \in S} \nabla_\theta \ell(z_i, \theta^*).$$

Substituting into the expansion, we have:

$$0 \approx 0 + \left(\frac{1}{n-m} \sum_{z_i \in S} \nabla_\theta \ell(z_i, \theta^*)\right) \epsilon + \mathbf{H}\, \Delta_\epsilon.$$

Solving for $\Delta_\epsilon$:

$$\Delta_\epsilon = -\mathbf{H}^{-1} \left[\frac{1}{n-m} \sum_{z_i \in S} \nabla_\theta \ell(z_i, \theta^*)\right] \epsilon.$$

Thus:

$$\left.\frac{d\,\theta^*_{\epsilon,S}}{d\,\epsilon}\right|_{\epsilon=0} = -\frac{1}{n-m} \mathbf{H}^{-1} \sum_{z_i \in S} \nabla_\theta \ell(z_i, \theta^*).$$

Applying the chain rule to measure the influence of the group $S$ on the loss at a test point $x$:

$$\left.\frac{d\,\ell(x, \theta^*_{\epsilon,S})}{d\,\epsilon}\right|_{\epsilon=0} = \nabla_\theta \ell(x, \theta^*)^\top \left.\frac{d\,\theta^*_{\epsilon,S}}{d\,\epsilon}\right|_{\epsilon=0}.$$

Substituting the expression for $\frac{d\,\theta^*_{\epsilon,S}}{d\,\epsilon}$, we obtain the first-order group influence function:

$$f_{\text{IF}}(x, S) = -c^{(1)} \nabla_\theta \ell(x, \theta^*)^\top \mathbf{H}^{-1} \sum_{z_i \in S} \nabla_\theta \ell(z_i, \theta^*),$$

where $c^{(1)} = \frac{1}{n-m}$.

### B.3  Deriving the Second-Order Group Influence Function

Since $\theta^*_{\epsilon,S}$ minimizes $L_S$, its gradient vanishes:

$$0 = \nabla_\theta L_S(\theta^*_{\epsilon,S}).$$

We perform a Taylor expansion around $\epsilon = 0$ and $\theta = \theta^*$, keeping terms up to $\epsilon^2$. Let:

$$\Delta_\epsilon = \Delta_1\, \epsilon + \Delta_2\, \epsilon^2 + O(\epsilon^3).$$

**First-Order $\epsilon$**   At $\epsilon = 0$, $L_S(\theta)$ reduces to $L(\theta)$, so:

$$\nabla_\theta L_S(\theta^*)\big|_{\epsilon=0} = \nabla_\theta L(\theta^*) = 0.$$

For the linear term ($\epsilon^1$), we have:

$$0 \approx \left.\frac{\partial}{\partial \epsilon} \nabla_\theta L_S(\theta^*, \epsilon)\right|_{\epsilon=0} + \nabla_\theta^2 L_S(\theta^*, 0)\Delta_1.$$

Hence:

$$\Delta_1 = -\mathbf{H}^{-1} \frac{1}{n-m} \sum_{z_i \in S} \nabla_\theta \ell(z_i, \theta^*).$$

This result matches the first-order group influence function.

**Second-Order $\epsilon$**   For the quadratic term ($\epsilon^2$), expanding the gradient $\nabla_\theta L_S$ and collecting all $\epsilon^2$ contributions yields:

$$0 \approx \frac{1}{2} \left.\frac{\partial^2}{\partial\epsilon^2}\nabla_\theta L_S(\theta^*)\right|_{\epsilon=0} + \left.\frac{\partial}{\partial\epsilon}\nabla_\theta^2 L_S(\theta^*)\right|_{\epsilon=0} \Delta_1$$
$$+ \mathbf{H}\,\Delta_2 + \frac{1}{2}\Delta_1^\top\left[\nabla_\theta^3 L(\theta^*)\right]\Delta_1.$$

Solving for $\Delta_2$:

$$\Delta_2 = - \mathbf{H}^{-1}\left[\tfrac{1}{2}\left.\tfrac{\partial^2}{\partial\epsilon^2}\nabla_\theta L_S(\theta^*)\right|_{\epsilon=0}\right.$$
$$+ \left.\tfrac{\partial}{\partial\epsilon}\nabla_\theta^2 L_S(\theta^*)\right|_{\epsilon=0}\Delta_1$$
$$+ \left.\tfrac{1}{2}\Delta_1^\top\nabla_\theta^3 L(\theta^*)\Delta_1\right].$$

Since the weights of the training examples are linear in $\epsilon$, the second derivative with respect to $\epsilon$ of these weights is zero:

$$\frac{1}{2}\left.\frac{\partial^2}{\partial\epsilon^2}\nabla_\theta L_S(\theta^*)\right|_{\epsilon=0} = 0.$$

Assuming higher-order terms $\nabla_\theta^k \ell(z;\theta)$ for $k \geq 3$ are negligible, the $\nabla_\theta^3 L(\theta^*)$ term vanishes. Thus:

$$\Delta_2 = -\mathbf{H}^{-1}\left.\frac{\partial}{\partial\epsilon}\nabla_\theta^2 L_S(\theta^*)\right|_{\epsilon=0}\Delta_1.$$

From the definition of $L_S$, we have:

$$\Delta_2 = -\mathbf{H}^{-1}\left[\frac{1}{n}\left(\sum_{z_i\in S}\nabla_\theta^2\ell(z_i,\theta^*)\right.\right.$$
$$\left.\left. - \frac{m}{n-m}\sum_{z_i\notin S}\nabla_\theta^2\ell(z_i,\theta^*)\right)\Delta_1\right]$$

Rearranging:

$$\Delta_2 = \frac{m}{n-m}\left(I - \mathbf{H}^{-1}\mathbf{H}_S\right)\Delta_1,$$

where $\mathbf{H}_S = \frac{1}{m}\sum_{z_i\in S}\nabla_\theta^2\ell(z_i,\theta^*)$.

**Influence on the Test Point**   For a test point $x$, consider $\ell\big(x,\theta_{\epsilon,S}^*\big)$. We aim to expand it as a Taylor series in $\theta$ around $\theta^*$. Following previous studies, we perform a first-order expansion rather than a second-order one. This choice is justified because the dominant second-order effects can typically be captured through the parameter shift $\Delta_\epsilon$. Thus, we obtain:

$$\ell\big(x,\theta_{\epsilon,S}^*\big) \approx \ell(x,\theta^*) + \nabla_\theta\ell(x,\theta^*)^\top\Delta_1\,\epsilon$$
$$+ \nabla_\theta\ell(x,\theta^*)^\top\Delta_2\,\epsilon^2.$$

The term multiplying $\epsilon$ corresponds to the first-order group influence, while the term multiplying $\epsilon^2$ is the second-order correction. In closed form:

**First-Order Group Influence:**

$$f_{\text{IF}}^{(1)}(x,S) = \nabla_\theta\ell\big(x,\theta^*\big)^\top\Delta_1,$$

**Second-Order Influence:**

$$f_{\text{IF}}^{(2)}(x,S) = \nabla_\theta\ell\big(x,\theta^*\big)^\top\Delta_1 + \nabla_\theta\ell\big(x,\theta^*\big)^\top\Delta_2$$

where:

$$\Delta_2 = \frac{m}{n-m}\left(I - \mathbf{H}^{-1}\mathbf{H}_S\right)\Delta_1,$$
$$\Delta_1 = -\mathbf{H}^{-1}\frac{1}{n-m}\sum_{z_i\in S}\nabla_\theta\ell(z_i,\theta^*).$$

**Simplifying Terms** Now clean up the expressions $f_{\text{IF}}^{(2)}(x, S)$. First, define:

$$\phi(z) = \mathbf{H}^{-\frac{1}{2}} \nabla_\theta \ell(z, \theta^*),$$

where $\phi(z)$ represents the preconditioned gradient of the loss.

Rearranging $f_{\text{IF}}^{(2)}(x, S)$, we get:

$$f_{\text{IF}}^{(2)}(x, S) = \underbrace{\nabla_\theta \ell(x, \theta^*)^\top \Delta_1 + \frac{m}{n-m} \ell(x, \theta^*)^\top \Delta_1}_{\text{Term } A}$$

$$\underbrace{- \frac{m}{n-m} \ell(x, \theta^*) \mathbf{H}^{-1} \mathbf{H}_S \Delta_1}_{\text{Term } B}$$

where terms $A$ and $B$ emerge.

Substituting $\phi(z)$ in $A$, we have:

$$A = -\frac{n}{n-m} \phi(x) \sum_{z_i \in S} \phi(z_i).$$

For term $B$, we have:

$$B = -\nabla_\theta \ell(x, \theta^*)^\top \frac{m}{n-m} \mathbf{H}^{-1} \mathbf{H}_S \Delta_1.$$

Approximating the subset Hessian $\mathbf{H}_S$ using the Fisher Information Matrix (FIM):

$$\mathbf{H}_S = \sum_{z_i \in S} \nabla_\theta^2 \ell(z_i, \theta^*) \approx \sum_{z_i \in S} \nabla_\theta \ell(z_i, \theta^*) \nabla_\theta \ell(z_i, \theta^*)^\top.$$

Substituting this approximation into $B$, we have:

$$B \approx \frac{m}{(n-m)^2} \phi(x) \left( \sum_{z_i \in S} \phi(z_i) \phi(z_i)^\top \right) \left( \sum_{z_i \in S} \phi(z_i) \right).$$

For further simplification, we define:

$$\alpha(z_i, S) = \phi(z_i)^\top \sum_{z_j \in S} \phi(z_j),$$

which measures the alignment of $\phi(z_i)$ with the aggregated gradients in $S$. Then:

$$B \approx \frac{m}{(n-m)^2} \phi(x) \sum_{z_i \in S} \left[ \alpha(z_i, S) \phi(z_i) \right].$$

Putting together, we get:

$$\begin{aligned}
f_{\text{IF}}^{(2)}(x, S) &= A + B \\
&= c^{(1)} \phi(x) \sum_{z_i \in S} \phi(z_i) + c^{(2)} \phi(x) \sum_{z_i \in S} \left[ \alpha(z_i, S) \phi(z_i) \right] \\
&= \phi(x) \sum_{z_i \in S} \left[ c^{(1)} + c^{(2)} \alpha(z_i, S) \right] \phi(z_i) \\
&= \phi(x) \sum_{z_i \in S} \sum_{t=1}^{2} \left[ c^{(t)} \alpha_t(z_i, S) \right] \phi(z_i)
\end{aligned}$$

where $c^{(1)}$ and $c^{(2)}$ are the constants of the first- and second-order terms, defined as:

$$c^{(1)} = \frac{n}{n-m},$$
$$c^{(2)} = \frac{m}{(n-m)^2}.$$

Additionally,

$$\alpha_1(z_i, S) = 1,$$
$$\alpha_2(z_i, S) = \phi(z_i)^\top \sum_{z_j \in S} \phi(z_j)$$

are the first- and second-order alignment functions.

## B.4 Deriving the Third-Order Group Influence Function

For many practical purposes, second-order corrections already capture the leading "beyond first-order" effects. However, in order to have a more complete perception of the structure of the influence function (and also to address our curiosity), we will continue to derive the third-order group influence function.

Since $\theta^*_{\epsilon,S}$ minimizes $L_S$, its gradient vanishes:

$$0 = \nabla_\theta L_S(\theta^*_{\epsilon,S}).$$

We perform a Taylor expansion around $\epsilon = 0$ and $\theta = \theta^*$, keeping terms up to $\epsilon^3$. Let:

$$\Delta_\epsilon = \Delta_1\,\epsilon + \Delta_2\,\epsilon^2 + \Delta_3\,\epsilon^3 + O(\epsilon^4).$$

The first-order and second-order terms, $\Delta_1$ and $\Delta_2$, have been derived in the previous section. Now focus on the third-order term, $\Delta_3$.

Considering the $\epsilon^3$ term and collecting all contributions of order $\epsilon^3$, we obtain:

$$0 \approx \underbrace{\frac{1}{6}\left.\frac{\partial^3}{\partial \epsilon^3}\nabla_\theta L_S(\theta^*)\right|_{\epsilon=0}}_{(i)} + \underbrace{\nabla^2_\theta L_S(\theta^*, 0)\,\Delta_3}_{(ii)}$$

$$+ \underbrace{\frac{1}{2}\left.\frac{\partial^2}{\partial \epsilon^2}\nabla^2_\theta L_S(\theta^*, 0)\right|_{\epsilon=0}\Delta_1}_{(iii)}$$

$$+ \underbrace{\left.\frac{\partial}{\partial \epsilon}\nabla^2_\theta L_S(\theta^*, 0)\right|_{\epsilon=0}\Delta_2}_{(iv)}$$

$$+ \underbrace{\Delta_1^\top \nabla^3_\theta L_S(\theta^*, 0)\,\Delta_2}_{(v)}$$

$$+ \underbrace{\frac{1}{2}\Delta_1^\top\left.\frac{\partial}{\partial \epsilon}\nabla^3_\theta L_S(\theta^*, 0)\right|_{\epsilon=0}\Delta_1}_{(vi)}.$$

Now analyze these terms in turn:

- Term (i) vanishes since the per-sample weight depend linearly on $\epsilon$:

$$(i) = \left.\frac{\partial^3}{\partial \epsilon^3}\nabla_\theta L_S(\theta^*)\right|_{\epsilon=0} = 0.$$

- Term (ii) can be written as:

$$(ii) = \nabla^2_\theta L_S(\theta^*, 0)\,\Delta_3 = \mathbf{H}\,\Delta_3.$$

- Term (iii) vanishes because the per-sample weight is linear in $\epsilon$:

$$(iii) = \frac{1}{2}\left.\frac{\partial^2}{\partial \epsilon^2}\nabla^2_\theta L_S(\theta^*, 0)\right|_{\epsilon=0}\Delta_1 = 0.$$

- Term (iv) is:

$$(\text{iv}) = \left. \frac{\partial}{\partial \epsilon} \nabla_\theta^2 L_S(\theta^*, 0) \right|_{\epsilon=0} \Delta_2.$$

Similar to the previous derivation, this can be rearranged as:

$$(\text{iv}) = \frac{m}{n-m} \left( \mathbf{H} - \mathbf{H}_S \right) \Delta_2,$$

where $\mathbf{H}_S = \frac{1}{m} \sum_{z_i \in S} \nabla_\theta^2 \ell(z_i, \theta^*)$.

- Term (v) involves the third derivative with respect to $\theta$. In most influence-function deriva-tions, one often assumes that higher-order derivatives $\nabla_\theta^3 L(\theta^*)$ are negligible. Following this convention, we consider:

$$(\text{v}) = \Delta_1^\top \nabla_\theta^3 L_S(\theta^*, 0) \, \Delta_2 = 0.$$

- Term (vi) is the derivative with respect to $\epsilon$ of the third derivative with respect to $\theta$. Since we neglect $\nabla_\theta^3 \ell$, any derivative of it also vanishes. Thus, we have:

$$(\text{vi}) = \frac{1}{2} \Delta_1^\top \left. \frac{\partial}{\partial \epsilon} \nabla_\theta^3 L_S(\theta^*, 0) \right|_{\epsilon=0} \Delta_1 = 0.$$

Putting everything together, the $\epsilon^3$ stationarity condition simplifies to

$$0 \approx \mathbf{H} \, \Delta_3 + \frac{m}{n-m} \left( \mathbf{H} - \mathbf{H}_S \right) \Delta_2.$$

Solving for $\Delta_3$ yields

$$\Delta_3 = -\frac{m}{n-m} \left( I - \mathbf{H}^{-1} \mathbf{H}_S \right) \Delta_2.$$

**Influence on the Test Point** For a test point $x$, consider $\ell(x, \theta^*_{\epsilon,S})$. Similar to the previous section, we can express the effects of $S$ on $x$ (when considering the $\epsilon^3$ term) as:

$$f_{\text{IF}}^{(3)}(x, S) = \nabla_\theta \ell(x, \theta^*)^\top \left( \Delta_1 + \Delta_2 + \Delta_3 \right).$$

The terms related to $\Delta_1$ and $\Delta_2$ have already been simplified in our derivation of the second-order group influence function, and we can write them as the following simplified expression:

$$f_{\text{IF}}^{(2)}(x, S) = \underbrace{c^{(1)} \phi(x) \sum_{z_i \in S} \phi(z_i)}_{\text{Term } A} + \underbrace{c^{(2)} \phi(x) \sum_{z_i \in S} \left[ \alpha(z_i, S) \phi(z_i) \right]}_{\text{Term } B}. \tag{11}$$

Therefore, the first two terms $A$ and $B$ of $f_{\text{IF}}^{(3)}(x, S)$ share the same structure, with constants slightly adjusted due to the inclusion of additional correction terms. Moreover, $f_{\text{IF}}^{(3)}(x, S)$ will have an additional term $C$ for the $\epsilon^3$ term. Let:

$$f_{\text{IF}}^{(3)}(x, S) = A + B + C.$$

Now, lets simplify $C$:

$$C = \left( \mathbf{H}^{-1} \mathbf{H}_S \right)^2 \mathbf{H}^{-1} \sum_{z_i \in S} \nabla_\theta \ell(z_i, \theta^*).$$

Approximating the subset Hessian $\mathbf{H}_S$ using the Fisher Information Matrix (FIM):

$$\mathbf{H}_S = \sum_{z_i \in S} \nabla_\theta^2 \ell(z_i, \theta^*) \approx \sum_{z_i \in S} \nabla_\theta \ell(z_i, \theta^*) \nabla_\theta \ell(z_i, \theta^*)^\top.$$

Substituting this approximation into $C$, we obtain:

$$C \approx c^{(3)} \phi(x) \left( \sum_{z_i \in S} \phi(z_i) \phi(z_i)^\top \right)^2 \left( \sum_{z_i \in S} \phi(z_i) \right).$$

For further simplification, we define:

$$\beta(z_i, S) \; = \; \phi(z_i)^\top \sum_{z_j \in S} \big[\alpha(z_j, S)\, \phi(z_j)\big],$$

where $\alpha(z_i, S)$ is the same as the previous definition, that is:

$$\alpha(z_i, S) \; = \; \phi(z_i)^\top \sum_{z_j \in S} \phi(z_j).$$

Then, we can express $C$ as:

$$C \; \approx \; c^{(3)}\, \phi(x) \sum_{z_i \in S} \Big[\beta(z_i, S)\, \phi(z_i)\Big].$$

Combining this with $A$ and $B$, we obtain:

$$\begin{aligned}
f_{\mathrm{IF}}^{(3)}(x, S) &= A + B + C \\
&= c^{(1)}\phi(x) \sum_{z_i \in S} \phi(z_i) \\
&\quad + c^{(2)}\phi(x) \sum_{z_i \in S} \big[\alpha(z_i, S)\phi(z_i)\big] \\
&\quad + c^{(3)}\, \phi(x) \sum_{z_i \in S} \Big[\beta(z_i, S)\, \phi(z_i)\Big].
\end{aligned}$$

We can then rename $\alpha$, $\beta$, and the constants to obtain:

$$f_{\mathrm{IF}}^{(3)}(x, S) = \phi(x) \sum_{z_i \in S} \sum_{t=1}^{3} \big[c_t^{(3)}\alpha_t(z_i, S)\big]\phi(z_i),$$

where $c_t^{(3)}$ are constants defined by the distributional properties of $S$, and

$$\begin{aligned}
\alpha_1(z_i, S) &= 1, \\
\alpha_2(z_i, S) &= \phi(z_i)^\top \sum_{z_j \in S} \phi(z_j), \\
\alpha_3(z_i, S) &= \phi(z_i)^\top \sum_{z_j \in S} \alpha_2(z_j, S)\phi(z_j).
\end{aligned}$$

These are the alignment functions, which can also be defined recursively as:

$$\alpha_t(z) = \begin{cases} 1, & \text{if } t = 1, \\ \phi(z)^\top \sum_{z_j \in S} \alpha_{t-1}(z_j)\phi(z_j), & \text{if } t \geq 2. \end{cases}$$

This gives us the same format as Theorem B.1.

## B.5 Deriving the Higher-Order Group Influence Function

We extend the previous derivation to higher-order terms. Before manually doing this, lets first review the effective terms (i.e., nonzero terms) of the first, second, and third-order $\epsilon$ terms:

For first order:

$$0 \approx \frac{\partial}{\partial \epsilon} \nabla_\theta L_S(\theta^*, \epsilon)\bigg|_{\epsilon=0} + \nabla_\theta^2 L_S(\theta^*, 0)\Delta_1.$$

For second order:

$$0 \approx \frac{\partial}{\partial \epsilon} \nabla_\theta^2 L_S(\theta^*, \epsilon)\bigg|_{\epsilon=0} \Delta_1 + \nabla_\theta^2 L_S(\theta^*, 0)\Delta_2.$$

For third order:

$$0 \approx \frac{\partial}{\partial \epsilon} \nabla_\theta^2 L_S(\theta^*, \epsilon)\Big|_{\epsilon=0} \Delta_2 + \nabla_\theta^2 L_S(\theta^*, 0)\Delta_3.$$

We see that at each order $k$ in $\epsilon$, the only "new" term that enters the equations (and thus the recursion for $\Delta_k$) is the first derivative with respect to $\epsilon$ multiplied by lower-order $\Delta_j$'s, plus the usual expansions in powers of $\Delta_\epsilon$ (where $\Delta_\epsilon = \theta^*_{\epsilon,S} - \theta^*$).

To explain this, we start with the stationarity condition at all $\epsilon$:

$$0 = \nabla_\theta L_S(\theta^* + \Delta_\epsilon, \epsilon).$$

Define:

$$G(\theta, \epsilon) := \nabla_\theta L_S(\theta, \epsilon).$$

Since $\theta^*$ is the minimizer at $\epsilon = 0$, we know $G(\theta^*, 0) = 0$.

Expanding $G(\theta^* + \Delta_\epsilon, \epsilon)$ using a multi-variable Taylor series and keeping terms up to first order in $\Delta_\epsilon$ gives:

$$G(\theta^* + \Delta_\epsilon, \epsilon) = \sum_{p=0}^{\infty} \sum_{r=0}^{\infty} \frac{1}{p!r!} \left[ \frac{\partial^{p+r} G}{\partial \epsilon^p \partial \theta^r}(\theta^*, 0) \right] [\Delta_\epsilon^r]\epsilon^p,$$

$$\approx G(\theta^*, 0) + \nabla_\theta^2 L_S(\theta^*, 0)\Delta_\epsilon$$

$$+ \sum_{p=1}^{\infty} \frac{1}{p!} \frac{\partial^p}{\partial \epsilon^p} G(\theta^*, 0)\epsilon^p$$

$$+ \sum_{p=1}^{\infty} \frac{1}{p!} \frac{\partial^p}{\partial \epsilon^p} \left[ \nabla_\theta^2 L_S(\theta^*, 0) \right] \epsilon^p \Delta_\epsilon.$$

Define:

$$\mathbf{H} = \nabla_\theta^2 L_S(\theta^*, 0), \quad \mathbf{H}' = \frac{\partial}{\partial \epsilon} \nabla_\theta^2 L_S(\theta, \epsilon)\Big|_{\theta=\theta^*, \epsilon=0}.$$

Because each training sample's weight depends linearly on $\epsilon$, we have:

$$\frac{\partial^p}{\partial \epsilon^p}\left[ \text{weights} \right] = 0 \quad \text{for } p \geq 2.$$

Thus, higher-order terms in $\frac{\partial^2}{\partial \epsilon^2} \nabla_\theta L_S$, $\frac{\partial^3}{\partial \epsilon^3} \nabla_\theta L_S$, etc., vanish, simplifying the stationarity expansions. In other words:

$$\frac{\partial^p}{\partial \epsilon^p} G(\theta^*, 0)\Big|_{p \geq 2} = 0 \quad \text{and} \quad \frac{\partial^p}{\partial \epsilon^p} \nabla_\theta^2 L_S(\theta^*, 0)\Big|_{p \geq 2} = 0.$$

Furthermore, collecting the $\epsilon^k$ terms in the stationarity condition $G(\theta + \Delta_\epsilon, \epsilon) = 0$ for $k \geq 1$ gives:

$$0 = \mathbf{H}\,\Delta_k + \mathbf{H}'\,\Delta_{k-1}.$$

Since $\Delta_0 = 0$ by definition, and $\Delta_1$ is determined by $\left[ \frac{\partial}{\partial \epsilon} G(\theta, 0) \right]$, we solve for:

$$\Delta_k = -\mathbf{H}^{-1}\mathbf{H}'\Delta_{k-1}.$$

By repeated substitution, we obtain closed-form expressions for all $\Delta_k$:

- Order 1:

$$\Delta_1 = -\mathbf{H}^{-1}\left[ \frac{\partial}{\partial \epsilon} G(\theta^*, 0) \right].$$

- Order $k \geq 2$:

$$\Delta_k = -\mathbf{H}^{-1}\mathbf{H}'\Delta_{k-1}.$$

This leads to the recursively defined alignment functions in the final group influence function formula. This completes the proof of Theorem B.1 for all cases where $k > 3$.

## C  Comparison between gradient-based and representation-based methods

From Eq. 3 and Eq. 5, it is evident that gradient-based and representation-based methods share a similar structure: both rely on the inner product of two vectors and differ only in terms of the calculation of these vectors. Specifically, the former utilizes gradients and and Hessian, while the latter uses embeddings. Additionally, under typical assumptions both classes of methods produce group influence estimates via the same aggregation process (i.e. simply summing per-example influence).

Apart from computational considerations, what factors might influence the choice between a gradient- or representation-based data attribution method? We highlight three considerations:

- **Accuracy**: whether the attribution results are reliable. Gradient-based methods are designed to simulate the model optimization process, which are theoretically accurate under certain assumptions. However, existing representation-based methods rely on heuristic metrics that are agnostic to the objective of data attribution, leading to sub-optimal results.

- **Computation cost**: Gradient-based methods generally rely on the computation of gradients and some approximation of the Hessian, making them computationally expensive. In practice, the cost of gradient-based methods is comparable to training the model on the all of the target examples because computing influence involves computing loss gradients for each target example. In contrast, representation-based methods are efficient and applicable for large-scale retrieval [66].

- **Storage cost**: Influence functions can be particularly storage-intensive as they can involve storing the full gradient (equal in size to the model parameters themselves) for each data sample, which prevents large-scale data attribution. Follow-up work projects gradients with random matrices [22] or low-rank approximations [25] at the cost of reduced accuracy. Representation-based methods output fixed-size embeddings, which are more manageable in size.

In summary, representation-based data attribution methods are cheaper than gradient-based method but do not reflect the actual process of model optimization.

## D  Details of Evaluation Datasets

We use four datasets for evaluation:

- **FLAN:** We randomly sampled 100,000 examples from the original FLAN training set as the TDA evaluation training split. For the test set, we retained the first 100 examples of each task in the test set. Since there are 66 tasks, the test set contains 6,520 examples.

- **Alpaca:** We use the original Alpaca training set (https://huggingface.co/datasets/tatsu-lab/alpaca) as the training data. For the test data, we design the test set to consist of two parts: (i) a seen subset, where we randomly sample 250 instructions from the training set and use a Llama model trained on the full Alpaca training set to generate the responses; (ii) an unseen subset, where we sample 250 examples from https://huggingface.co/datasets/tatsu-lab/alpaca_eval. Thus, the total test set consists of 500 examples.

- **Tulu:** We use version v1 of Tulu (https://huggingface.co/datasets/allenai/tulu-v1-sft-mixture). The original dataset contains over 490k instances. We randomly sample 100,000 examples for our training set and 500 for our test set.

- **SafeRLHF:** SafeRLHF (https://huggingface.co/datasets/PKU-Alignment/PKU-SafeRLHF-QA) is a dataset for safety alignment in large language models, where each example has a safety label of either "safe" or one of 19 possible harm categories. We use the original training set. For the test set, we sample 500 examples from the original test set, ensuring a 250:250 ratio of safe and harmful instances for label balance.

See Table 3 for statistics of evaluation datasets.

## E  Details of Baselines

Our baselines include two categories: gradient-based methods and representation-based methods.

Table 3: Evaluation Data for data attribution. # Train and # Test denotes number of examples in training and testing set.

| Name | # Train | # Test |
|------|---------|--------|
| FLAN | 100,000 | 6,520 |
| Alpaca | 51,760 | 500 |
| Tulu | 100,000 | 500 |
| SafeRLHF | 251,963 | 500 |

Gradient-based methods rely on the gradient of data with respect to a warmed-up model. To warm up the model, for FLAN, we fine-tune it on 100,000 examples randomly sampled from the FLAN training set. For Alpaca, Tulu, and SafeRLHF, we warm up the model on 100,000 examples randomly sampled from UltraChat.

**LoGra**    LoGra is a toolkit for efficient influence function calculation (`https://github.com/logix-project/logix`), and its core techniques are introduced in Choe et al. [16]. Its key idea is to use low-rank gradient projection to achieve implicit gradient projection during backpropagation, thereby reducing computational cost. We experimented with different configurations and found that the following variants achieved the best performance in our evaluation (as also described in Eq. 3): (i) applying the LoGra module to all MLP layers, (ii) initializing the LoGra weights using PCA, (iii) approximating the Hessian using KFAC, and (iv) performing unit normalization of vectors.

The rank value of LoGra controls the projection dimension. We set it to 4, 8, and 16, resulting in a final dimension ranging from 1,152 to 27,648.

**TracIn**    TracIn [22] is derived from a first-order approximation of model weights and essentially computes the dot product between gradients without Hessian correction. For implementation efficiency, we use LoGra to perform gradient projection and apply unit normalization to the vectors. The original TracIn keeps multiple checkpoints and ensembles their influence scores, but this approach linearly increases computation and storage costs. Instead, we use only one checkpoint (i.e., the final checkpoint) of the warmed-up LM.

**LESS**    LESS [25] adopts LoRA tuning instead of full fine-tuning and projects the LoRA gradient to a lower dimension using a random matrix. Its key idea is to use AdamW states (first- and second-order momentum) to correct the gradients. We use the official implementation (`https://github.com/princeton-nlp/LESS`) and set the projection dimensions to 768 and 8192, which are the default dimensions in the LESS paper.

For LM warm-up, we follow their paper by setting the LoRA rank to 128 and alpha to 512, fine-tuning for 4 epochs. Similar to TracIn, the original LESS method ensembles influence scores from 4 checkpoints. For a fair comparison, we only retain the final checkpoint.

Another note is that the official implementation does not support batched gradient calculation, requiring a batch size of 1, which slows down execution. To better utilize the GPU, we run multiple processes on a single GPU to accelerate computation and report the computation time. However, its speed remains lower than LoGra, which supports batched computation.

**DsDm and TRAK**    TRAK [12] is a representative gradient-based method that provides an efficient implementation of the influence function. TRAK approximates the Hessian using the Fisher Information Matrix (FIM), ensembles influence scores across multiple checkpoints, and adjusts the influence score of each data sample by multiplying it with the model's prediction probability. DsDm [33] later applied TRAK to LLM data selection tasks. Since DsDm has not released its official code (`https://github.com/MadryLab/DsDm`), we approximate its implementation (i.e., TRAK) using the Logix framework, where the Hessian is approximated with FIM (also known as raw Hessian approximation in Logix), LoGra is used to project the gradient to different dimensions with random weight initialization, and only the final checkpoint is applied without checkpoint ensembling.

The following are representation-based baselines.

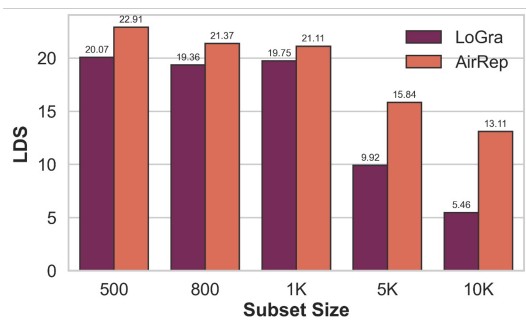

Figure 8: LDS performance of LoGra and AirRep under different evaluation subset sizes. AirRep consistently outperforms LoGra, especially on larger data size. Note that AirRep is trained only on data with a subset size of 1K.

**TF-IDF**    We use *sklearn* to calculate the cosine similarity of TF-IDF feature of data.

**DSIR**    DSIR utilizes hashed n-gram features, where n-grams are mapped onto a fixed number of buckets for efficiency and scalability. The influence estimator is parameterized by a bag-of-words generative model on the hashed n-grams, learned simply by counting the hash bucket frequencies. We use the official implementation (https://github.com/p-lambda/dsir).

**RDS**    RDS was introduced in Hanawa et al. [6], which uses the model's hidden states as data representations. Through an evaluation of different representation design choices, we use average pooling over the last-layer hidden states of all tokens as the sentence embedding.

**GTE**    GTE [35] is a general text embedding model trained on a mixture of text retrieval, text classification, text clustering, and text entailment data. It achieves strong performance on text embedding leaderboards relative to its model size (e.g., https://huggingface.co/spaces/mteb/leaderboard). For direct evaluation, we assessed various GTE model sizes, including Small, Base, Large, 1.5B, and 7B.

The following are discussion about other TDA approaches which we did considered for empirical comparison.

**Datamodels**    AirRep is inspired by Datamodels [1], in terms of problem formulation for data attribution and both methods focus on estimating empirical group influence scores. A key difference lies in how embeddings are generated. Datamodels assign fixed embedding vectors for all instances in a given data collection, and require the full retraining when new instances are added. AirRep, on the other hand, uses a trained encoder to produce the embeddings for any new instance on the fly. As a result, AirRep is computationally much more efficient than Datamodels in handling dynamically changing data. Empirical findings in [12] shows that Datamodels typically incur inference costs around 100x greater than gradient-based methods to achieve comparable performance. AirRep addresses this issue with the trained encoder, and being 80x more efficient when handling new instances. We will add these discussions in our revised version of the paper.

# F    Training Cost

For data generation, we create 10K subsets (each containing 1K data examples) and train the model on each subset for 2 epochs. This results in a total of 10M training examples for the Qwen2.5-0.5B LM, requiring about 20 hours on eight A100 GPUs. During the optimization stage, each training step involves sampling 32 subsets (1K examples each), totaling 32K examples per gradient descent step. The model is trained for up to 2K steps, completing in about 5 hours on an eight-GPU machine. While this training cost exceeds the warm-up training requirements of gradient-based TDA methods, it is exchanged for significant efficiency gains during inference. See Section 9 for detailed analysis.

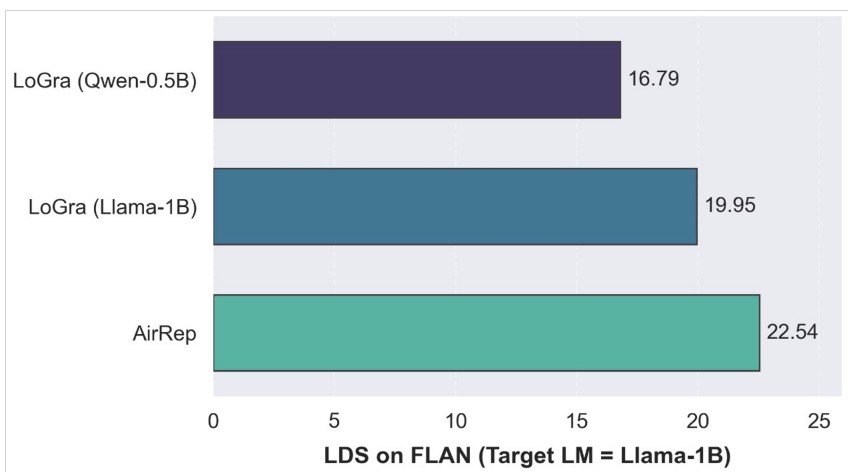

Figure 9: LDS scores of LoGra and AirRep when the target LM is LLaMA-1B. For LoGra, we evaluate both variants: LoGra applied to Qwen-0.5B and LoGra applied to LLaMA-1B. AirRep is trained on Qwen-0.5B data and directly evaluated using LLaMA-1B labels. Despite the model shift, AirRep achieves strong LDS performance, even outperforming LoGra applied to LLaMA-1B, demonstrating its ability to generalize across similar LMs without re-training.

Table 4: Training Cost of AirRep. Asymptotic complexity and wall-clock runtime (measured as eight A100 GPU hours) for each training stage. $n$ denotes the size of each subset, $N_s$ represents the total number of subsets, $E$ is the number of epochs for training on each subset, $M$ is the number of sampled subsets per training step, and $T$ is the total number of training steps for AirRep.

| Stage | Data Generation | Optimization |
|---|---|---|
| Complexity | $\mathcal{O}(n \cdot N_s \cdot E)$ | $\mathcal{O}(n \cdot M \cdot T)$ |
| Actual Value | $\mathcal{O}(1K \cdot 10K \cdot 2)$ | $\mathcal{O}(1K \cdot 32 \cdot 2K)$ |

## G Data Selection Setup

Our data selection evaluation is conducted independently for each task in FLAN, using a greedy search strategy to identify the most relevant training data. For a given task, the test set contains $m$ examples (typically $m = 100$ due to our data sampling strategies), denoted as $(x_1, x_2, \ldots, x_m)$. Let the training set be $(z_1, z_2, \ldots, z_n)$. The TDA method computes an influence score $f(x_i, z_j)$ for each pair $(x_i, z_j)$, where $i \in [m]$ and $j \in [n]$. Next, we determine the rank of each training example relative to a given test example. Specifically, we define $\text{rank}(z_i, x_j) \in [n]$ as the position of $x_j$ among the most influential training examples for $z_i$. In other words, if $\text{rank}(z_i, x_j) = 1$, then $x_j$ is the most influential training example for $z_i$. We then define the score of each test example $x_j$ as the best (lowest) rank it achieves across all training examples: $\text{Score}(x_j) = \min(\text{rank}(z_i, x_j) \mid i \in [m])$. This represents the highest influence rank of $x_j$ among all test examples. Finally, we rank the training data based on $\text{Score}(x_j)$ and select the top 1,000 examples.

After training the target LM on the selected data, we evaluate the model on the test set, setting the maximum generation length to 64. We use unigram F1 as the evaluation metric for all tasks, even though some tasks have other conventional metrics, such as accuracy for classification, ROUGE for summarization, and BLEU for translation. Notably, unigram F1 is correlated with these task-specific metrics since the model's generations are typically very short. For example, in sentiment classification task, the model is expected to generate a single token (*positive* or *negative*), making unigram F1 equivalent to accuracy. Finally, the final F1 score is averaged over 66 tasks in FLAN.

For gradient-based methods like LESS and LoGra, which can be configured to balance computational and storage costs, we use a 0.5B LM as the proxy model to compute gradients. We set the projection dimensions to 768 and 1152 for LESS and LoGra, respectively, as this ensures a comparable storage requirement to AirRep (which uses a 384-dimensional representation) while remaining a practical configuration for data selection from large-scale corpora

# H Additional Evaluation

## H.1 LOO Evaluation

To analyze the effectiveness of our method on individual data influence estimation, we conducted a leave-one-out (LOO) evaluation, in which individual examples were removed and we measured the correlation between predicted and actual influence scores. The results on a FLAN subset are summarized in Table 5. We find that AIRREP outperforms baseline methods in this setting as well.

Table 5: LOO correlation between predicted and actual influence scores on a FLAN subset.

| Method | LOO Correlation |
|---|---|
| LoGra (18432) | 10.75 |
| GTE-Small | 0.72 |
| AirRep | **15.36** |

## H.2 Data Classification Results

Table 6: Accuracy of Data Classification.

| Method | FLAN | Tulu | SafeRLHF |
|---|---|---|---|
| LoGra (Dim 1152) | 71.61 | 76.60 | 78.00 |
| LoGra (Dim 18432) | 85.44 | 86.00 | 83.20 |
| DsDm (Dim 18432) | 83.68 | 85.34 | 80.25 |
| TracIn (Dim 18432) | 77.69 | 79.43 | 67.12 |
| GTE-Small (Dim 384) | 50.59 | 76.60 | **90.60** |
| AirRep (Dim 384) | **86.41** | **88.20** | 87.20 |

# I Detailed Data Selection Results

See Table 7, 8, 9 for detailed results of data selection.

Table 7: Data Selection Results for Qwen2.5-0.5B LM.

| Data | Random | DSIR | TF-IDF | GTE | LESS | LoGra | AirRep |
|---|---|---|---|---|---|---|---|
| aeslc | 21.64 | 16.86 | 26.61 | 27.17 | 25.53 | 27.2 | 29.61 |
| ag_news_subset | 76.86 | 67.61 | 79.41 | 74.75 | 82.85 | 82.97 | 77.72 |
| anli_r1 | 28.14 | 28.02 | 38.9 | 36.04 | 50.44 | 40.13 | 44.49 |
| anli_r2 | 34.02 | 29.7 | 40.17 | 32.91 | 35.01 | 41.96 | 36.58 |
| anli_r3 | 36.55 | 24.47 | 35.55 | 37.54 | 36.7 | 36.17 | 41.98 |
| arc_challenge | 37.15 | 36.84 | 42.47 | 41.33 | 41.76 | 45.07 | 39.15 |
| arc_easy | 46.38 | 43.27 | 43.71 | 48.45 | 51.89 | 48.86 | 48.8 |
| bool_q | 49.0 | 48.0 | 54.0 | 53.0 | 56.0 | 55.0 | 56.0 |
| cb | 12.7 | 31.81 | 38.73 | 34.64 | 56.67 | 70.6 | 56.88 |
| cnn_dailymail | 19.68 | 12.65 | 24.19 | 23.09 | 22.83 | 21.76 | 26.99 |
| cola | 57.23 | 62.18 | 63.24 | 63.18 | 61.48 | 63.18 | 63.43 |
| common_gen | 41.69 | 45.23 | 45.78 | 43.2 | 46.26 | 43.41 | 45.1 |
| copa | 60.29 | 60.59 | 61.79 | 64.53 | 63.94 | 64.54 | 66.34 |
| coqa | 10.99 | 6.36 | 39.21 | 28.72 | 37.42 | 41.98 | 41.86 |
| cosmos_qa | 39.02 | 40.01 | 39.92 | 42.04 | 43.17 | 43.15 | 39.54 |
| dart | 62.73 | 66.65 | 66.08 | 68.98 | 67.42 | 67.39 | 67.7 |
| definite_pronoun | 46.07 | 52.73 | 51.0 | 58.0 | 57.07 | 54.67 | 54.07 |
| drop | 8.08 | 13.49 | 10.69 | 13.18 | 19.35 | 20.0 | 14.81 |
| e2e_nlg | 63.48 | 68.12 | 67.91 | 65.83 | 66.34 | 68.38 | 65.82 |
| fix_punct | 93.43 | 95.08 | 95.17 | 95.44 | 94.93 | 95.2 | 95.24 |
| gigaword | 26.59 | 26.13 | 28.46 | 27.14 | 28.68 | 28.06 | 29.73 |
| glue_mrpc | 55.0 | 69.0 | 51.0 | 57.0 | 66.0 | 68.0 | 73.0 |
| glue_qqp | 66.0 | 72.0 | 78.0 | 74.0 | 78.0 | 71.0 | 79.0 |
| hellaswag | 29.31 | 28.89 | 37.21 | 38.62 | 45.81 | 46.02 | 43.3 |
| imdb_reviews | 58.57 | 21.88 | 64.85 | 65.76 | 70.43 | 69.39 | 70.41 |
| math_dataset | 3.0 | 2.0 | 7.0 | 6.0 | 3.0 | 1.0 | 4.0 |
| mnli_matched | 67.0 | 79.0 | 85.0 | 63.0 | 81.0 | 81.0 | 84.0 |
| mnli_mismatched | 77.0 | 73.0 | 74.0 | 74.0 | 81.0 | 76.0 | 78.0 |
| multi_news | 15.45 | 3.38 | 19.57 | 18.56 | 10.98 | 19.47 | 19.24 |
| multirc | 11.06 | 0.34 | 37.26 | 51.0 | 36.0 | 29.0 | 36.0 |
| natural_questions | 6.61 | 8.57 | 10.41 | 7.04 | 7.43 | 7.79 | 6.77 |
| openbookqa | 48.61 | 37.93 | 48.11 | 49.52 | 52.74 | 50.22 | 54.65 |
| opinion_abstracts_idebate | 11.13 | 9.91 | 17.85 | 20.44 | 20.1 | 18.82 | 21.31 |
| opinion_abstracts_rotten | 6.97 | 4.33 | 11.62 | 14.06 | 9.9 | 8.51 | 14.01 |
| para_crawl_enes | 46.02 | 43.37 | 48.66 | 47.28 | 45.21 | 46.23 | 46.03 |
| paws_wiki | 53.0 | 51.0 | 61.0 | 63.0 | 83.0 | 91.0 | 83.0 |
| piqa | 74.34 | 76.46 | 77.71 | 77.91 | 76.49 | 79.53 | 78.45 |
| qnli | 68.92 | 44.49 | 68.98 | 60.51 | 74.58 | 78.81 | 71.17 |
| quac | 4.72 | 15.15 | 19.34 | 17.57 | 22.26 | 28.23 | 18.75 |
| record | 18.38 | 16.42 | 18.38 | 16.45 | 17.55 | 16.1 | 10.8 |
| rte | 64.08 | 47.21 | 56.13 | 67.78 | 57.06 | 52.74 | 65.81 |
| samsum | 28.03 | 13.63 | 34.45 | 34.95 | 34.29 | 35.19 | 36.85 |
| sentiment140 | 48.35 | 40.95 | 33.68 | 40.36 | 49.02 | 47.8 | 39.9 |
| snli | 77.1 | 77.94 | 76.21 | 78.24 | 80.1 | 79.94 | 77.14 |
| squad_v1 | 28.33 | 26.42 | 34.24 | 38.07 | 39.19 | 37.87 | 40.24 |
| squad_v2 | 33.24 | 11.64 | 36.8 | 24.75 | 50.5 | 56.0 | 48.8 |
| sst2 | 69.13 | 68.0 | 70.23 | 71.39 | 69.27 | 71.61 | 69.76 |
| story_cloze | 64.36 | 57.71 | 64.47 | 67.62 | 78.17 | 81.92 | 77.51 |
| stsb | 25.74 | 32.24 | 29.54 | 35.92 | 34.52 | 33.26 | 41.0 |
| trec | 48.24 | 79.0 | 75.89 | 63.39 | 78.34 | 75.25 | 72.41 |
| trivia_qa | 9.05 | 10.49 | 9.87 | 10.96 | 10.97 | 12.52 | 9.27 |
| true_case | 96.64 | 97.28 | 98.71 | 98.07 | 98.98 | 99.43 | 99.22 |
| web_nlg_en | 70.25 | 73.5 | 72.7 | 72.72 | 71.46 | 74.95 | 76.62 |
| wic | 55.0 | 56.0 | 52.0 | 44.0 | 49.0 | 55.0 | 40.0 |
| wiki_lingua_english_en | 15.99 | 11.87 | 18.56 | 17.9 | 19.54 | 16.91 | 20.95 |
| wmt14_enfr | 46.07 | 37.69 | 44.19 | 46.46 | 46.74 | 46.78 | 47.17 |
| wmt16_translate_csen | 24.58 | 21.91 | 24.57 | 24.75 | 26.8 | 25.78 | 27.68 |
| wmt16_translate_deen | 46.25 | 43.36 | 46.4 | 46.2 | 45.22 | 45.39 | 46.03 |
| wmt16_translate_fien | 22.54 | 19.79 | 24.18 | 22.13 | 22.93 | 22.61 | 24.72 |
| wmt16_translate_roen | 34.01 | 30.65 | 34.38 | 36.4 | 32.56 | 34.17 | 35.65 |
| wmt16_translate_ruen | 39.96 | 35.99 | 40.59 | 39.13 | 41.4 | 40.67 | 40.11 |
| wmt16_translate_tren | 29.4 | 24.42 | 26.85 | 28.88 | 25.61 | 27.44 | 29.5 |
| wnli | 57.14 | 57.14 | 61.43 | 58.57 | 58.57 | 54.29 | 57.14 |
| word_segment | 71.65 | 87.1 | 82.68 | 80.29 | 83.79 | 85.93 | 91.07 |
| wsc | 65.0 | 59.0 | 61.0 | 50.0 | 63.0 | 65.0 | 65.0 |
| yelp_polarity_reviews | 72.95 | 63.22 | 73.24 | 73.66 | 73.53 | 71.81 | 74.46 |

Table 8: Data Selection Results for Qwen2.5-1.5B LM.

| Data | Random | DSIR | TF-IDF | GTE | LESS | LoGra | AirRep |
|---|---|---|---|---|---|---|---|
| aeslc | 21.59 | 20.66 | 28.66 | 28.56 | 27.0 | 25.96 | 27.05 |
| ag_news_subset | 79.47 | 79.37 | 77.22 | 80.21 | 82.15 | 80.5 | 82.27 |
| anli_r1 | 44.93 | 44.46 | 48.75 | 50.67 | 51.04 | 59.91 | 54.44 |
| anli_r2 | 47.12 | 45.21 | 44.7 | 37.04 | 42.31 | 48.8 | 53.12 |
| anli_r3 | 39.98 | 36.8 | 40.05 | 43.03 | 44.8 | 42.91 | 43.76 |
| arc_challenge | 49.39 | 52.05 | 52.44 | 52.9 | 57.32 | 56.7 | 58.1 |
| arc_easy | 59.02 | 61.85 | 58.46 | 64.19 | 60.65 | 63.54 | 64.6 |
| bool_q | 60.0 | 48.0 | 63.0 | 62.0 | 74.0 | 78.0 | 72.0 |
| cb | 30.86 | 48.2 | 29.15 | 71.11 | 77.04 | 53.02 | 75.22 |
| cnn_dailymail | 18.45 | 9.19 | 28.8 | 27.6 | 27.29 | 27.97 | 27.69 |
| cola | 50.29 | 72.07 | 63.32 | 70.17 | 73.17 | 64.15 | 45.32 |
| common_gen | 47.56 | 50.01 | 50.08 | 49.44 | 48.26 | 51.34 | 47.91 |
| copa | 71.68 | 73.35 | 75.61 | 74.29 | 74.53 | 73.37 | 72.45 |
| coqa | 9.78 | 4.89 | 31.64 | 26.47 | 36.81 | 32.24 | 42.75 |
| cosmos_qa | 46.48 | 44.01 | 45.56 | 49.96 | 50.01 | 49.81 | 49.89 |
| dart | 65.46 | 66.73 | 68.79 | 69.85 | 66.71 | 68.57 | 68.29 |
| definite_pronoun | 50.89 | 50.0 | 53.67 | 52.0 | 59.0 | 52.0 | 57.67 |
| drop | 14.24 | 16.78 | 17.8 | 21.75 | 23.23 | 22.89 | 20.68 |
| e2e_nlg | 66.71 | 67.82 | 68.02 | 67.35 | 68.41 | 67.05 | 67.78 |
| fix_punct | 94.48 | 94.98 | 95.32 | 95.52 | 95.39 | 96.25 | 95.69 |
| gigaword | 29.49 | 27.74 | 29.92 | 28.86 | 30.04 | 29.36 | 31.02 |
| glue_mrpc | 49.0 | 70.0 | 76.0 | 68.0 | 73.0 | 58.0 | 67.0 |
| glue_qqp | 78.0 | 83.0 | 84.0 | 78.0 | 81.0 | 81.0 | 78.0 |
| hellaswag | 45.13 | 29.01 | 49.58 | 54.93 | 64.47 | 58.89 | 60.07 |
| imdb_reviews | 59.78 | 28.44 | 71.69 | 68.64 | 67.19 | 66.67 | 66.76 |
| math_dataset | 3.0 | 3.0 | 8.0 | 3.0 | 1.0 | 3.0 | 4.0 |
| mnli_matched | 79.0 | 84.0 | 67.0 | 86.0 | 80.0 | 90.0 | 86.0 |
| mnli_mismatched | 84.0 | 83.0 | 79.0 | 86.0 | 84.0 | 84.0 | 83.0 |
| multi_news | 12.63 | 4.33 | 19.39 | 19.27 | 11.65 | 19.68 | 19.69 |
| multirc | 22.7 | 1.37 | 27.45 | 23.48 | 46.0 | 53.0 | 48.0 |
| natural_questions | 11.99 | 14.75 | 16.47 | 15.17 | 11.82 | 14.43 | 13.86 |
| openbookqa | 57.04 | 55.77 | 56.37 | 55.42 | 61.75 | 62.42 | 58.44 |
| opinion_abstracts_idebate | 13.78 | 12.47 | 19.48 | 22.28 | 20.55 | 21.71 | 23.96 |
| opinion_abstracts_rotten | 6.08 | 5.25 | 13.84 | 13.05 | 11.69 | 10.86 | 15.03 |
| para_crawl_enes | 51.87 | 52.19 | 53.94 | 52.89 | 52.9 | 52.94 | 53.59 |
| paws_wiki | 62.0 | 71.0 | 77.0 | 82.0 | 87.0 | 86.0 | 92.0 |
| piqa | 77.96 | 77.9 | 81.91 | 80.38 | 80.03 | 81.1 | 80.32 |
| qnli | 77.9 | 46.08 | 61.87 | 76.73 | 78.8 | 78.08 | 79.22 |
| quac | 10.77 | 14.75 | 13.78 | 19.75 | 27.78 | 26.69 | 17.78 |
| record | 15.65 | 14.58 | 13.45 | 15.16 | 15.67 | 12.53 | 10.62 |
| rte | 70.93 | 63.51 | 66.81 | 71.88 | 71.02 | 62.85 | 58.81 |
| samsum | 34.09 | 21.79 | 40.03 | 40.6 | 38.11 | 38.89 | 39.88 |
| sentiment140 | 42.67 | 46.46 | 47.87 | 49.02 | 48.57 | 50.72 | 42.49 |
| snli | 83.1 | 83.46 | 86.58 | 66.09 | 83.46 | 82.94 | 76.17 |
| squad_v1 | 44.56 | 43.91 | 45.59 | 48.18 | 48.3 | 46.95 | 51.21 |
| squad_v2 | 39.54 | 22.24 | 43.42 | 41.07 | 59.0 | 58.9 | 61.06 |
| sst2 | 70.83 | 72.22 | 70.32 | 71.79 | 74.05 | 75.64 | 74.07 |
| story_cloze | 73.85 | 79.2 | 82.17 | 84.04 | 85.49 | 86.29 | 86.13 |
| stsb | 26.88 | 43.59 | 37.17 | 35.78 | 29.36 | 25.6 | 41.1 |
| trec | 62.52 | 82.03 | 82.03 | 77.41 | 78.89 | 79.67 | 76.41 |
| trivia_qa | 29.45 | 19.97 | 18.42 | 22.13 | 21.0 | 19.28 | 20.09 |
| true_case | 98.75 | 98.99 | 98.88 | 99.5 | 99.75 | 99.73 | 99.75 |
| web_nlg_en | 72.41 | 76.42 | 76.89 | 77.02 | 75.69 | 76.39 | 76.65 |
| wic | 57.0 | 43.0 | 47.0 | 63.0 | 45.0 | 46.0 | 56.0 |
| wiki_lingua_english_en | 18.37 | 12.41 | 19.16 | 20.22 | 20.49 | 16.25 | 19.87 |
| wmt14_enfr | 55.94 | 51.51 | 55.44 | 53.22 | 55.23 | 55.22 | 55.78 |
| wmt16_translate_csen | 35.5 | 33.26 | 36.68 | 35.76 | 37.13 | 35.96 | 39.24 |
| wmt16_translate_deen | 54.86 | 51.96 | 54.89 | 55.12 | 55.78 | 53.42 | 53.88 |
| wmt16_translate_fien | 29.23 | 27.37 | 30.5 | 30.86 | 27.53 | 30.09 | 31.11 |
| wmt16_translate_roen | 41.61 | 37.97 | 44.19 | 41.22 | 41.87 | 42.78 | 42.24 |
| wmt16_translate_ruen | 47.36 | 45.48 | 45.56 | 48.7 | 45.91 | 47.82 | 49.07 |
| wmt16_translate_tren | 38.49 | 34.42 | 38.3 | 40.47 | 38.33 | 36.59 | 38.79 |
| wnli | 62.86 | 57.14 | 67.14 | 58.57 | 44.29 | 50.0 | 67.14 |
| word_segment | 82.06 | 90.72 | 90.31 | 90.78 | 87.07 | 89.64 | 90.36 |
| wsc | 65.0 | 35.0 | 65.0 | 36.0 | 65.0 | 42.0 | 57.0 |
| yelp_polarity_reviews | 74.94 | 66.41 | 77.04 | 62.37 | 76.02 | 76.68 | 74.79 |

Table 9: Data Selection Results for Qwen2.5-3B LM.

| Data | Random | DSIR | TF-IDF | GTE | LESS | LoGra | AirRep |
|---|---|---|---|---|---|---|---|
| aeslc | 22.94 | 25.58 | 31.06 | 30.76 | 26.32 | 28.1 | 32.1 |
| ag_news_subset | 82.79 | 49.79 | 75.93 | 78.6 | 80.28 | 79.08 | 79.92 |
| anli_r1 | 56.4 | 43.82 | 57.49 | 52.64 | 45.6 | 62.64 | 55.25 |
| anli_r2 | 37.08 | 39.16 | 52.42 | 36.01 | 46.08 | 44.18 | 39.18 |
| anli_r3 | 38.95 | 44.1 | 43.07 | 39.65 | 44.86 | 58.84 | 48.98 |
| arc_challenge | 62.1 | 60.91 | 63.18 | 64.28 | 65.41 | 69.1 | 64.43 |
| arc_easy | 60.46 | 64.55 | 66.64 | 66.44 | 67.47 | 64.67 | 66.84 |
| bool_q | 72.0 | 64.0 | 76.0 | 66.0 | 66.0 | 76.0 | 76.0 |
| cb | 36.95 | 68.78 | 73.76 | 70.97 | 68.8 | 67.18 | 76.89 |
| cnn_dailymail | 21.55 | 8.77 | 29.1 | 27.4 | 15.03 | 26.56 | 29.74 |
| cola | 63.34 | 63.0 | 71.0 | 52.13 | 43.46 | 78.58 | 79.38 |
| common_gen | 49.08 | 47.42 | 48.97 | 48.44 | 47.24 | 48.72 | 48.36 |
| copa | 73.81 | 68.18 | 72.25 | 70.86 | 76.38 | 74.87 | 77.18 |
| coqa | 5.75 | 2.51 | 31.11 | 24.45 | 27.37 | 43.23 | 43.33 |
| cosmos_qa | 53.96 | 51.83 | 54.15 | 52.9 | 54.28 | 56.99 | 54.0 |
| dart | 65.09 | 66.37 | 69.82 | 70.53 | 69.55 | 69.06 | 70.21 |
| definite_pronoun | 52.0 | 56.0 | 52.0 | 55.0 | 49.0 | 53.0 | 63.0 |
| drop | 15.29 | 21.8 | 20.24 | 21.74 | 27.99 | 22.72 | 22.09 |
| e2e_nlg | 64.11 | 66.03 | 67.87 | 65.88 | 66.93 | 66.71 | 66.38 |
| fix_punct | 95.08 | 94.46 | 95.88 | 95.4 | 95.94 | 95.57 | 95.76 |
| gigaword | 28.77 | 27.11 | 31.32 | 31.37 | 29.66 | 29.92 | 32.01 |
| glue_mrpc | 70.0 | 74.0 | 53.0 | 32.0 | 79.0 | 72.0 | 50.0 |
| glue_qqp | 79.0 | 79.0 | 75.0 | 78.0 | 79.0 | 76.0 | 80.0 |
| hellaswag | 57.54 | 49.18 | 61.75 | 56.11 | 64.21 | 68.58 | 69.16 |
| imdb_reviews | 67.74 | 27.28 | 64.5 | 63.8 | 72.9 | 70.13 | 71.74 |
| math_dataset | 5.33 | 6.0 | 8.0 | 5.0 | 5.0 | 7.0 | 5.0 |
| mnli_matched | 76.0 | 89.0 | 78.0 | 87.0 | 87.0 | 92.0 | 64.0 |
| mnli_mismatched | 87.0 | 89.0 | 67.0 | 87.0 | 89.0 | 87.0 | 85.0 |
| multi_news | 6.42 | 3.29 | 20.25 | 20.29 | 8.2 | 20.33 | 20.26 |
| multirc | 24.08 | 2.94 | 45.53 | 52.0 | 54.0 | 54.0 | 46.46 |
| natural_questions | 16.98 | 19.6 | 15.28 | 15.28 | 18.21 | 17.0 | 17.78 |
| openbookqa | 61.94 | 62.07 | 60.43 | 63.8 | 62.54 | 64.34 | 60.08 |
| opinion_abstracts_idebate | 14.6 | 10.67 | 22.65 | 21.71 | 19.8 | 20.59 | 23.67 |
| opinion_abstracts_rotten | 9.23 | 6.16 | 11.94 | 16.47 | 12.85 | 11.53 | 17.36 |
| para_crawl_enes | 54.92 | 56.77 | 55.78 | 55.94 | 55.08 | 55.51 | 55.55 |
| paws_wiki | 69.0 | 76.0 | 54.0 | 81.0 | 92.0 | 87.0 | 89.0 |
| piqa | 78.8 | 74.89 | 78.67 | 79.4 | 81.72 | 79.95 | 80.78 |
| qnli | 71.33 | 75.33 | 78.95 | 66.37 | 79.61 | 70.57 | 77.84 |
| quac | 13.29 | 11.81 | 17.9 | 27.32 | 29.35 | 26.82 | 27.23 |
| record | 19.74 | 17.4 | 19.5 | 17.01 | 15.33 | 14.88 | 16.39 |
| rte | 54.7 | 66.22 | 59.02 | 50.07 | 68.13 | 64.91 | 67.95 |
| samsum | 32.46 | 21.29 | 40.79 | 41.16 | 38.62 | 38.67 | 42.4 |
| sentiment140 | 47.71 | 45.72 | 48.66 | 45.45 | 48.52 | 50.44 | 38.4 |
| snli | 74.1 | 59.36 | 85.34 | 80.79 | 81.94 | 72.16 | 83.1 |
| squad_v1 | 40.29 | 44.99 | 41.19 | 48.93 | 51.69 | 45.93 | 54.25 |
| squad_v2 | 38.4 | 26.21 | 38.43 | 37.14 | 59.77 | 54.82 | 61.38 |
| sst2 | 70.33 | 65.0 | 73.63 | 74.87 | 73.48 | 73.65 | 72.93 |
| story_cloze | 82.93 | 76.82 | 84.51 | 85.94 | 84.14 | 86.27 | 86.43 |
| stsb | 37.82 | 35.56 | 38.11 | 33.6 | 29.03 | 42.23 | 37.22 |
| trec | 64.12 | 80.41 | 57.66 | 74.58 | 85.07 | 79.85 | 80.41 |
| trivia_qa | 43.57 | 31.93 | 35.72 | 28.77 | 31.63 | 30.28 | 26.53 |
| true_case | 97.48 | 99.22 | 98.99 | 99.21 | 99.45 | 99.41 | 99.38 |
| web_nlg_en | 75.18 | 78.21 | 78.76 | 77.45 | 75.37 | 77.94 | 80.07 |
| wic | 56.0 | 44.0 | 54.0 | 44.0 | 44.0 | 42.0 | 44.0 |
| wiki_lingua_english_en | 17.6 | 10.1 | 19.76 | 19.78 | 17.7 | 16.1 | 19.82 |
| wmt14_enfr | 55.52 | 55.11 | 57.02 | 57.87 | 57.51 | 58.27 | 57.11 |
| wmt16_translate_csen | 41.69 | 38.36 | 41.63 | 40.6 | 41.45 | 39.79 | 43.68 |
| wmt16_translate_deen | 55.85 | 55.17 | 56.59 | 56.47 | 55.14 | 56.85 | 57.45 |
| wmt16_translate_fien | 31.53 | 31.17 | 33.66 | 34.73 | 32.27 | 33.07 | 32.54 |
| wmt16_translate_roen | 46.34 | 43.67 | 47.0 | 45.59 | 42.55 | 45.64 | 45.28 |
| wmt16_translate_ruen | 51.89 | 48.51 | 52.63 | 50.32 | 52.01 | 52.94 | 51.71 |
| wmt16_translate_tren | 40.43 | 42.03 | 43.06 | 41.96 | 42.4 | 43.57 | 42.85 |
| wnli | 55.71 | 58.57 | 45.71 | 65.71 | 70.0 | 61.43 | 47.14 |
| word_segment | 85.99 | 90.12 | 93.99 | 89.57 | 92.07 | 93.04 | 92.66 |
| wsc | 66.0 | 65.0 | 58.0 | 36.0 | 54.0 | 35.0 | 65.0 |
| yelp_polarity_reviews | 69.79 | 62.21 | 76.74 | 74.89 | 74.93 | 74.1 | 76.97 |

