# OpenReview forum: "Enhancing Training Data Attribution with Representational Optimization"
_NeurIPS.cc/2025/Conference — NeurIPS 2025 spotlight_

### Official Review · Reviewer_32FH · 2025-05-31

**Clarity:** 3
**Significance:** 2
**Originality:** 2
**Rating:** 4
**Confidence:** 3

**Summary:**

This paper introduces AirRep, a training data attribution (TDA) method to explain how the training data contribute to the model prediction. Previous TDA methods, such as gradient-based methods Tracin and LoGra, have theoretical grounds, its high computational cost hinders scalability for LLMs. Existing representation-based TDA is more efficient but relies on heuristic design and the representations are not optimized for attribution, limiting accuracy.
AirRep adopts a trainable encoder and attention-based pooling to learn task-specific and model-aligned representations explicitly for TDA, where the core idea is to accurately group influence estimation. Experiments demonstrate AirRep's performance is comparable to computationally expensive gradient methods.

**Questions:**

- AirRep uses BERT for the encoder. Why BERT?
- What exactly is the "re-training outcome" in line 83
- My understanding is that this AirRep approach learns task-specific and model-aligned representations explicitly optimised for TDA. Therefore, it is not efficient that for each downstream task, we need to train again? Although previous TDAs are also task-dependent, but AirRep is quite expensive to train compared to representation-base TDA. Is it correct?
- This paper only considers fine-tuning, where he knowledge bias present in the base model before fine-tuning is ignored, Is this understanding correct?
- Is it possible to use KSD [3] as an evaluation metric?

**Ethical Concerns:**

["NO or VERY MINOR ethics concerns only"]

**Final Justification:**

Most of my concerns and misunderstandings have been addressed. Based on the comments and other reviewers' comments, I would like to increase my score.

**Limitations:**

Yes

**Quality:**

3

**Strengths And Weaknesses:**

**Strengths**

- Research on TDA is an important step towards understanding how LLMs learn from their training data, which is fundamental to ensure transparency and accountability in AI systems.
- AirRep achieves performance on par with the stoa gradient-based approaches. Specifically, it outperforms LoGra in the LDS evaluation.

**Weaknesses**

- The paper only uses the LDS evaluation as the sole metric for comparison. Other metrics like Leave-One-Out or AOPC value are used in previous work [1].
- The weighted pairwise ranking loss uses thresholds to clip weights based on the difference in ground-truth scores. The specific values used for these thresholds in the experiments are not explicitly stated in the provided text. Seems like there is some heuristic engineering work. Not sure if the thresholds are generalizable to other models or datasets.

**Others**

- There are loads of papers cited in their arxiv versions, not their published versions. See example below:

Mengzhou Xia, Mikel Artetxe, Chunting Zhou, Xi Victoria Lin, Ramakanth Pasunuru, Danqi Chen, Luke Zettlemoyer, and Veselin Stoyanov. 2023. Training Trajectories of Language Models Across Scales. In Proceedings of the 61st Annual Meeting of the Association for Computational Linguistics (Volume 1: Long Papers), pages 13711–13738, Toronto, Canada. Association for Computational Linguistics.

Basu, Samyadeep, Phil Pope, and Soheil Feizi. "Influence Functions in Deep Learning Are Fragile." In International Conference on Learning Representations.

- The paper would benefit from presenting a qualitative analysis, accompanied by concrete examples.

**Missing related work**

[1] NIPS2023, Nguyen, Elisa, Minjoon Seo, and Seong Joon Oh. A bayesian approach to analysing training data attribution in deep learning.

[2] ACL2024, Kangxi Wu, Liang Pang, Huawei Shen, and Xueqi Cheng. 2024. Enhancing Training Data Attribution for Large Language Models with Fitting Error Consideration.

[3] ICLR2025, Mahtab Sarvmaili, Hassan Sajjad, Ga Wu, Data-centric Prediction Explanation via Kernelized Stein Discrepancy

---

> ### Author Rebuttal · Authors · 2025-07-31
>
> We thank the reviewer for engaging with our work and for recognizing the importance of TDA for model transparency, as well as AirRep’s strong performance. We will address each of your points in detail below, with particular focus on clarifying our contributions to AirRep’s novelty, scalability, and performance as a principled and robust framework for TDA.
>
> ---
>
> > **The paper only uses the LDS evaluation as the sole metric for comparison. Other metrics like Leave-One-Out or AOPC value are used in previous work [1].**
>
> Thank you for pointing this out! While LDS has been a widely used benchmark in recent TDA research, we agree that evaluating attribution quality with alternative metrics adds valuable perspective. To this end, we conducted a Leave-One-Out (**LOO**) analysis, where we removed individual training examples and measured the Pearson correlation between the predicted influence scores and the actual change in test loss.
> The results on a FLAN subset are summarized below:
>
> | Method  | LOO Correlation |
> | - | - |
> | LoGra (18432) |  10.75  |
> | GTE-Small | 0.72  |
> | AirRep | 15.36 |
>
> AirRep achieves higher correlation than baseline methods, suggesting that its optimization-based approach provides more robust and reliable individual-level attribution.  We will include this analysis and discussion in the revised version of the paper.
>
> ---
>
> > **The weighted pairwise ranking loss uses thresholds to clip weights based on the difference in ground-truth scores. The specific values used for these thresholds in the experiments are not explicitly stated in the provided text. Seems like there is some heuristic engineering work. Not sure if the thresholds are generalizable to other models or datasets.**
>
> Thank you for raising this concern. The clipping thresholds were set to 0.1 (low) and 5.0 (high), as described in the AirRep training details in Section 4. These values were not extensively tuned and were selected based on simple, generalizable observations:
> - The **low threshold (0.1)** corresponds to approximately one standard deviations of the loss difference observed from repeated fine-tuning of language models on the same data, which we treat as typical variation due to training stochasticity.
> - The **high threshold (5.0)** was empirically chosen to cap rare but extreme differences, which we consider outliers and clip to prevent them from disproportionately influencing the learning process.
>
> Importantly, we applied the **same thresholds and training hyperparameters across all experiments**, and observed consistently stable and performant training. We will clarify these points in the revised version to ensure transparency and reproducibility.
>
> ---
>
> > **There are loads of papers cited in their arxiv versions, not their published versions.**
>
> Thank you for pointing out this issue! We will correct these citations in our final version.
>
> ---
>
> > **The paper would benefit from presenting a qualitative analysis, accompanied by concrete examples.**
>
> Thank you for the suggestion. We agree that including qualitative examples can provide deeper intuition into how AirRep assigns attribution scores. In the revised version, we will include detailed case studies and concrete examples to illustrate attribution behavior. Additionally, we will open-source our data and models to support transparency and enable further exploration by the community.
>
> ---
>
> > **AirRep uses BERT for the encoder. Why BERT?**
>
> Thank you for this question. We initialize AirRep with a BERT-style encoder (specifically, GTE-small) due to its strong performance in text embedding tasks and ease of training. GTE-small is well-suited for pairwise scoring due to its retrieval-focused pretraining. That said, AirRep is not tied to any specific encoder architecture. We expect that other text embedding models could also be integrated successfully.
>
> ---
>
> > **What exactly is the "re-training outcome" in line 83**
>
> The “re-training outcome” mentioned refers to r(x, S), defined in line 79 as the loss of the model on example x after training on dataset S. We recognize that this phrasing could cause confusion and will clarify this in our final version.
>
> ---
>
> > **My understanding is that this AirRep approach learns task-specific and model-aligned representations explicitly optimised for TDA. Therefore, it is not efficient that for each downstream task, we need to train again? Although previous TDAs are also task-dependent, but AirRep is quite expensive to train compared to representation-base TDA. Is it correct?**
>
> In our evaluation, we did not retrain AirRep for each downstream task. Once trained, AirRep learns model-aligned representations for predicting influence scores via loss change estimation and can be applied to various downstream TDA tasks, such as data selection and classification. While we observe promising generalization to unseen data, we acknowledge that performance may vary in domains with substantially different distributions—such as complex reasoning tasks where long COT models generate significantly longer thinking content. Exploring domain-adaptive training strategies is a promising direction for future work.
>
> Regarding training cost, **Section 9** provides a detailed comparison with baseline TDA methods. We show that although AirRep incurs a one-time training cost, it becomes cost-effective after approximately 5.6 GPU hours, and achieves up to 80× faster inference than gradient-based approaches. Since the trained model can be reused across tasks, this cost can be amortized in multi-task settings, offering both principled modeling and practical scalability.
>
> ---
>
> > **This paper only considers fine-tuning, where he knowledge bias present in the base model before fine-tuning is ignored, Is this understanding correct?**
>
> While our evaluation focuses on fine-tuned models, **we do not ignore the influence of the pretrained model's biases**. AirRep’s supervision signal is derived from **actual training outcomes**, meaning that any biases inherited from the base model and expressed during fine-tuning are naturally reflected in the measured loss changes. Thus, our attribution framework implicitly captures the combined effects of both pretraining and fine-tuning.
>
> ---
>
> > **Is it possible to use KSD [3] as an evaluation metric?**
>
> Thank you for highlighting this interesting work. After reviewing [3], we agree that KSD presents a promising alternative for evaluating attribution methods. However, adapting it to our current LLM training setup would require non-trivial adjustments, particularly since the original implementation was designed for image classification.
>
> That said, because [3] also evaluates methods like Influence Functions and TracIn, we believe KSD could potentially be extended to evaluate AirRep as well. We consider this an exciting direction for future work and will explore it in follow-up studies.
>
> ---
>
> **Thank you again for your thoughtful feedback—we hope these clarifications will resolve your concerns.**

---

> > ### Author Response · Authors · 2025-08-02
> >
> > Dear Reviewer 32FH,
> >
> > Thank you for taking the time to review our paper and for your constructive comments! We have submitted our rebuttal, which provides detailed responses to your concerns regarding additional evaluation metrics, clarification on the choice of clip weights, and your other questions and suggestions. In particular, we have included new evaluations using the LOO metric.
> >
> > We hope our responses effectively address your concerns. We would greatly appreciate it if you could confirm whether our response has resolved them. We are happy to engage further if you have any additional questions.
> >
> > Thank you again!

---

> > > ### Comment · Reviewer_32FH · 2025-08-05
> > >
> > > Thanks for your reply. Most of my concerns and misunderstandings have been addressed. Based on the comments and other reviewers' comments, I will increase my score.

---

### Official Review · Reviewer_zyXJ · 2025-06-29

**Clarity:** 3
**Significance:** 3
**Originality:** 4
**Rating:** 5
**Confidence:** 4

**Summary:**

The paper proposes an optimisation-based approach to computing training data attribution scores, where a model consisting of an encoding module and an aggregation module is trained to predict the attribution score of a subset of the training data on a specific model output. The model is trained with true attribution scores, which the authors define as the model's loss on a test sample where the model was finetuned on the subset.

**Questions:**

- How many subsets $r$ and therefore additional training cost is required to get a well-performing AirRep model? Does it depend on model and data size of the model one wishes to investigate with training data attribution?
- How well does AirRep work for smaller/larger subset sizes than the ones reported in the paper? TDA methods are known to struggle to capture attribution for individual samples due to the training noise [a, b, c], does an optimization-based approach alleviate this?

[a] Basu et al. (2020): Influence Functions in Deep Learning Are Fragile.

[b] Epifano et al. (2023): Revisiting the fragility of influence functions.

[c] Nguyen et al. (2023): A bayesian approach to analysing training data attribution in deep learning.

**Ethical Concerns:**

["NO or VERY MINOR ethics concerns only"]

**Final Justification:**

I gave the score of accept because this paper presents a promising alternative, scalable approach to training data attribution and presents a convincing evaluation. The authors were responsive and able to answer questions and engage in discussions, demonstrating their fluency in the field. They also provided additional results comparing with a ground truth recorded through counterfactual LOO experiments, showing that their optimization-based approach to TDA is valid.

**Limitations:**

Yes, shortly in the conclusion section.

**Paper Formatting Concerns:**

None, maybe just a request to slightly increase the font size of Figure 3 for better readability.

Nits and typos found:
- In Equation 2, the transpose of the loss gradient vector is missing
- line 64: exhibit -> exhibits
- line 83: estimates -> estimate
- line 134: support -> supports

**Quality:**

3

**Strengths And Weaknesses:**

Strengths:
- To train a model for predicting attribution scores is an innovative approach to training data attribution that could scale well for larger models and datasets.
- The evaluation strongly supports AirRep's performance and generalizability across different subsets.
- The authors present a connection between higher-order influence functions and the attention-based pooling approach of AirRep.

Weaknesses:
- The paper would improve if the authors added a small paragraph or some sentences of how the related works relate to their work.
- It is unclear whether training an AirRep model requires specific tuning - how easy will it be to implement this?

---

> ### Author Rebuttal · Authors · 2025-07-31
>
> We sincerely thank the reviewer for the thoughtful and encouraging feedback. We appreciate the recognition of AirRep’s originality in formulating training data attribution as an optimization problem, its strong empirical performance and generalizability, and the connection to higher-order influence functions via attention-based pooling. We respond to each of your comments and suggestions in detail below.
>
> ---
>
> > **It is unclear whether training an AirRep model requires specific tuning - how easy will it be to implement this?**
>
> Thank you for this question. In our experience, **AirRep is relatively easy to train and implement.** The training data is generated automatically by sampling random subsets from training corpus, without requiring any specialized design or manual curation. Training follows a standard contrastive learning setup and does not rely on extensive hyperparameter tuning.
>
> Specifically, we used a fixed learning rate of 1e-4 and a batch size of 32, selected to fit typical GPU memory constraints. These settings yielded stable training across all experiments. We have also shared our data generation and training code in the Supplementary Material, which we hope will make implementation straightforward for future users.
>
> ---
>
> > **How many subsets r and therefore additional training cost is required to get a well-performing AirRep model? Does it depend on model and data size of the model one wishes to investigate with training data attribution?**
>
> Thank you for this important question. In **Section 9**, we provide a quantitative analysis of how training set size impacts AirRep’s performance. In our experiments, training with **several hundred random subsets** over approximately 5 GPU hours was sufficient for AirRep to outperform LoGra across multiple tasks.
>
> We found that the required number of subsets depends more on the diversity of the data distribution than on the model or dataset size. For example, more diverse datasets like Tulu typically benefit from more training subsets, whereas less diverse datasets such as Alpaca require fewer. Notably, we did **not observe saturation** in performance when increasing the number of training subsets in our current experiments. This suggests that the training cost can be flexibly traded off against performance and inference efficiency, depending on available resources.
>
> ---
>
> > **How well does AirRep work for smaller/larger subset sizes than the ones reported in the paper?**
>
> Thank you for this thoughtful question. In **Figure 6**, we report results on the FLAN dataset across a range of subset sizes. Although **AirRep was trained only on subsets of size 1000**, it consistently outperforms LoGra across both smaller and larger subset sizes at test time. AirRep’s advantage becomes more pronounced with larger subsets, likely due to its attention-based pooling mechanism, which is designed to explicitly model group-level influence.
>
> ---
>
> > **TDA methods are known to struggle to capture attribution for individual samples due to the training noise [a, b, c], does an optimization-based approach alleviate this?**
>
> Thank you for this insightful question. Capturing attribution at the individual-sample level is indeed challenging. Since AirRep is trained on many sampled subsets, this may have an **averaging effect** similar to repeated training, helping to stabilize the outcomes. Additionally, our *importance reweighting objective* is designed to reduce the influence of training noise.
>
> In addition, to further assess AirRep's performance on single sample attribution, we evaluated it using a Leave-One-Out (LOO) metric. where we remove a single training example and measure the actual change in test loss, then compare it to the predicted influence score. The results on a FLAN subset are summarized below:
>
> | Method | LOO Correlation |
> | - | - |
> | LoGra (18432) |  10.75  |
> | GTE-Small | 0.72  |
> | AirRep | 15.36 |
>
> AirRep outperforms the baselines in this setting, suggesting that its group-level optimization framework can yield stable attribution at the individual-example level. We will include a more detailed discussion of this analysis in the revised paper.
>
> ---
>
> > **Typos …**
>
> Thank you for pointing out the typos!  We will correct these in the revised version of the paper.
>
> ---
>
> **Thank you again for your thoughtful feedback—we hope these clarifications will resolve your concerns.**

---

> > ### Comment · Reviewer_zyXJ · 2025-08-01
> >
> > I would like to thank the authors for the detailed response, and would like to follow up on some questions:
> >
> > > **We found that the required number of subsets depends more on the diversity of the data distribution than on the model or dataset size.**
> >
> > I believe this is a valuable insight. Do you mean diversity in terms of task diversity or other types, e.g. language?
> >
> > > **LOO Correlation**
> >
> > Thank you for running these additional experiments, could you clarify if this is Spearman rank correlation (and bounded in [-100, 100])? Assuming this is the case, the correlations are rather weak (~10 is not really considered a correlation), even though I acknowledge that AirRep is better than Logra.

---

> > > ### Author Response · Authors · 2025-08-02
> > >
> > > Dear Reviewer zyXJ
> > >
> > > Thanks for your reply!
> > >
> > > ---
> > >
> > > > **I believe this is a valuable insight. Do you mean diversity in terms of task diversity or other types, e.g. language?**
> > >
> > > **Yes, here we are referring to task diversity**. For example, the Tulu dataset is more diverse than Alpaca, as it includes a broader range of tasks and domains such as CoT, code, classical NLP, and chitchat, and it also includes the Alpaca. This increased diversity requires a wider range of knowledge and capabilities. We found that on complex, diverse datasets like Tulu, AirRep requires relatively more training data to match the performance of LoGra.
> > >
> > > ---
> > >
> > > > **Thank you for running these additional experiments, could you clarify if this is Spearman rank correlation (and bounded in [-100, 100])? Assuming this is the case, the correlations are rather weak (~10 is not really considered a correlation), even though I acknowledge that AirRep is better than Logra.**
> > >
> > > **Yes, we are using Spearman correlation, scaled by 100.** We agree that the absolute correlation scores appear low. This may be due to the inherent stochasticity of LLM training and the relatively shallow training setup (e.g., 2 epochs, learning rate of 2e-5), which makes the influence of individual data points noisier.
> > >
> > > **However, we found such correlation levels are common and still meaningful in LLM SFT settings.** For example, current TDA methods achieve LDS correlation scores around ~10 when evaluated on Qwen2.5-3B (Figure 2), yet can select a small subset of data (e.g., 1k out of 100k) that performs comparably to or better than training on the full dataset (see Figure 3, Qwen2.5-3B). This demonstrates that even modest correlation can translate into practical gains in downstream tasks. Moreover, the relative performance of these methods remains consistent across evaluation settings, despite differences in the absolute scale of correlation scores (e.g., Table 1, Figure 2, Figure 6).
> > >
> > > ---
> > >
> > > We appreciate your insights and would be happy to engage further if you have any additional questions or concerns.

---

> > ### Comment · Reviewer_zyXJ · 2025-08-04
> >
> > Dear Authors,
> >
> > Thank you for your reply. It makes sense that more task-diverse datasets require more data/larger subset size to achieve the same performance. And it is interesting that even though correlations are generally low in TDA methods (not just AirRep), there is still sufficient signal for subset selection... I have no further questions, thank you!

---

> > > ### Author Response · Authors · 2025-08-05
> > >
> > > We appreciate your valuable feedback on our paper and are glad to have addressed your concerns.

---

### Official Review · Reviewer_wX2D · 2025-06-29

**Clarity:** 3
**Significance:** 2
**Originality:** 3
**Rating:** 5
**Confidence:** 3

**Summary:**

This paper introduces a new approach to training data attribution called AirRep. AirRep combines the advantages of gradient-based approaches with the efficiency of representtation-based methods. It encodes the training data with embeddings and scores their influence using inner-product with the provided example using attention based pooling. AirRep’s attention-based pooling mechanism captures group effects. The model is optimized using a weighted pairwise ranking objective over automatically generated influence signals since the expected usage is not pure influence scores but relative ordering of training examples. Evaluation on LLM fine-tuning demonstrates the effectiveness and generalization of AirRep. Authors also note that there is increased cost of training the embedding model but it can be done in advance.

**Questions:**

- Why some baselines (such as TracIn) are not evaluated in some sections?
- Why wasn't TRAK evaluated as a baseline?
- Would certain set of encoding models work better than others?

I am voting borderline accept for now, but if the question about baselines are addressed I am willing to increase my score and vice versa.

**Ethical Concerns:**

["NO or VERY MINOR ethics concerns only"]

**Final Justification:**

This paper provides a new, solid approach to training data attribution-- AirRep. I think this paper makes a good addition to interpretability methods. Authors have addressed nearly all my comments leaving some for future work.

**Limitations:**

One additional point could have been discussed as to whether this is NLP-specific TDA method or applicable to other general domains.

Other than that, the limitations has been adequately addressed.

**Quality:**

3

**Strengths And Weaknesses:**

### Quality
This paper has the following strengths:
* It adds a clever attention-based pooling to represent input groups much better. It is also nice of authors to apply the same logic to some of the baselines to measure the impact.
* Applying sigmoid-based training data confidence gating is also a good idea.
* The evaluations show good improvement over baselines.
* Its baselines (will touch on this later) are reasonable.
* Good sections on Ablation Study and Amortizing Training Cost.

The paper could have been improved in these areas:
* While it is understandable that the authors can't possibly evaluate their approach with all the metrics, this reviewer wonders why the direction of evaluation has been done in reverse (adding influential training examples rather than removing training examples) compared to other papers.
* A good evaluation could have been performed with non-random subsets. An evaluation on fact-tracing, which could be a very popular use case of this work, could go a long way.
* One of the more fundamental issues is lack of some important works such as TRAK for comparison.
* It is not clear why each experimental section contains only a subset of the mentioned baselines. If there is a good reason for dropping an approach from evaluation, it should be explicitly mentioned.
* nit: While it has been used before by other papers, correlation with another metrics is still not fully accepted form of evaluation.

### Clarity
The paper is well written in general. A few comments:
* Introduction and Preliminaries are too long.
* LDS is introduced in [12] and not in [1].
* Section 5 is called Data Attribution, but it is measuring more of counterfactual correctness.
* very nitpicky comment: "We define the model’s “prediction” on x as the cross-entropy loss computed on x, given by $r(x, S)$ = $ℓ(x; θ_*)$" -- defining prediction as this causes a mental load for the rest of the paper for readability as one is too used to assume predictions are regular model outputs.

### Significance
This paper adds to the plethora of training data attribution methods. Based on the improvement on the baselines (and lack of some), and added computational cost, this paper will be useful for some.

### Originality
The paper has few interesting additions on TDA: weighted attention for group selection, confidence-based example selection and ranking-based training objective. None of these approaches are novel in isolation, but this paper demonstrates their clever application.

---

> ### Author Rebuttal · Authors · 2025-07-31
>
> Thank you for your thoughtful and constructive review—your feedback highlights the strengths of our method (e.g., attention-based pooling, confidence gating, and ablation clarity) while also providing valuable suggestions on evaluation direction, baseline completeness (e.g., TRAK and TracIn), and broader applicability, which we will carefully address to improve the paper. We respond to each of your points in detail below.
>
> ---
>
> >  **why the direction of evaluation has been done in reverse (adding influential training examples rather than removing training examples) compared to other papers.**
>
> Thank you for this insightful observation. We chose to focus on **adding** influential examples—rather than removing them—primarily due to computational efficiency. In removal-based evaluation (e.g., Leave-One-Out LOO), each example must be removed from the training set and the model’s performance must be re-evaluated, often requiring a full retraining per removal. In contrast, additive evaluation can be performed efficiently by training on a selected subset, which is far more efficient in practice.
>
> However, following your suggestion, we conducted a **LOO evaluation**, in which individual examples were removed and we measured the correlation between predicted and actual influence scores. The results on a FLAN subset are summarized below:
>
> |  Method | LOO Correlation |
> | - | - |
> | LoGra (18432) |  10.75  |
> | GTE-Small | 0.72  |
> | AirRep | 15.36 |
>
> We find that **AirRep outperforms baseline** methods in this setting as well. We will include this new analysis and discussion in the revised version of the paper.
>
> **For context,** our evaluation setup otherwise follows established precedents: for LDS, we follow [Datamodels] to evaluate on randomly sampled subsets (see Figure 6 for different subset size), and for data selection and classification tasks, our setup mirrors that of LESS and Hanawa et al.
>
> ---
>
> >  **A good evaluation could have been performed with non-random subsets. An evaluation on fact-tracing, which could be a very popular use case of this work, could go a long way.**
>
> Thank you for this thoughtful suggestion. We agree that fact tracing is a compelling and high-impact application of training data attribution. While time constraints during the rebuttal period may prevented us from conducting additional evaluations on fact-tracing dataset, we see this as a promising future direction. We will add a discussion of fact-tracing applications in the future work section of the revised paper and plan to explore corresponding evaluations in follow-up work.
>
> ---
>
> >  **One of the more fundamental issues is lack of some important works such as TRAK for comparison. Why wasn't TRAK evaluated as a baseline?**
>
> Thank you for raising this important point. In our experiments, we include **DsDm** [REF], an improved variant of **TRAK** specifically optimized for LLM-scale settings. In our evaluation, we found that DsDm performs comparably to LoGra and is consistently outperformed by AirRep across tasks. We will clarify the relationship between TRAK and DsDm in the revised version of the paper to avoid confusion and ensure proper attribution.
>
> [REF] DsDm: Model-Aware Dataset Selection with Datamodels
>
> ---
>
> >  **It is not clear why each experimental section contains only a subset of the mentioned baselines. If there is a good reason for dropping an approach from evaluation, it should be explicitly mentioned. Why some baselines (such as TracIn) are not evaluated in some sections?**
>
> Thank you for pointing this out. In Table 1, we present a comprehensive comparison across all baselines. In subsequent experimental sections, we focus on a subset of baselines—specifically the most competitive or directly relevant ones (e.g., LoGra, LESS, and GTE-small)—to ensure clarity and conserve space in the main text. Baselines such as TracIn, which consistently underperform relative to LoGra, were omitted from detailed comparisons for conciseness.
>
> That said, we understand that this selective presentation may have caused confusion. To address this, we will (1) include **complete evaluation** results for all baselines in the Appendix, and (2) **explicitly state our rationale** for baseline selection in each experimental section in the revised version of the paper.
>
> ---
>
> >  **The paper is well written in general. A few comments …**
>
> Thank you for your detailed and constructive suggestions for improving clarity. In response, we will streamline the Introduction and Preliminaries sections to improve readability and focus. We will also revise the title of Section 5 to more accurately reflect its content, explicitly distinguishing between data attribution and counterfactual correctness. Additionally, we will clarify our use of the term “prediction” to avoid confusion with its conventional meaning in model outputs.
>
> ---
>
> >  **Would certain set of encoding models work better than others?**
>
> Thank you for this interesting question. In our preliminary experiments, we compared **GTE-small** and **DeBERTa-small**, and observed that GTE-small achieved slightly better performance—likely due to its pre-training on text retrieval tasks, which aligns well with our pairwise comparison objective.
>
> While GTE-small provided strong overall results, we agree that broader exploration of encoding models is worthwhile. In particular, models like **BERTScore** or **ColBERT**, which support fine-grained token-to-token matching, may offer further improvements in attribution fidelity. We view this as a promising direction and plan to explore it in future work.
>
> ---
>
> >  **One additional point could have been discussed as to whether this is NLP-specific TDA method or applicable to other general domains.**
>
> While our current evaluation focuses on language tasks, the design of AirRep is **modality-agnostic**—it relies only on the availability of suitable encoders and the ability to estimate loss differences. In principle, this framework could be applied to other domains such as vision or multimodal settings.
>
> We will include this point in the discussion of limitations and future work in the revised paper and plan to explore cross-domain generalization in follow-up research.
>
> ---
>
> **Thank you again for your thoughtful feedback—we hope these clarifications will resolve your concerns.**

---

> > ### Author Response · Authors · 2025-08-02
> >
> > Dear Reviewer wX2D,
> >
> > Thank you for taking the time to review our paper and for your constructive comments! We have submitted our rebuttal, which provides detailed responses to your concerns regarding our evaluation setup, comparison to TRAK, and your suggestions on paper writing. In particular, **we have compared our approach to TRAK through its LLM-focused variant, DsDm**, and **included additional evaluations**.
> >
> > We hope our responses effectively address your concerns. We would greatly appreciate it if you could confirm whether our response has addressed your concerns. We are happy to engage further if you have any additional questions.
> >
> > Thank you again!

---

> > ### Comment · Reviewer_wX2D · 2025-08-03
> >
> > Thank you for addressing my question and concerns.
> >
> > I will increase my rating.

---

> > > ### Author Response · Authors · 2025-08-05
> > >
> > > We appreciate your valuable feedback on our paper and are glad to have addressed your concerns.

---

### Official Review · Reviewer_VLz2 · 2025-07-04

**Clarity:** 4
**Significance:** 3
**Originality:** 4
**Rating:** 5
**Confidence:** 4

**Summary:**

The paper proposes AirRep, a representation-based training-data attribution (TDA) method for large language models (LLMs). AirRep preserves the bilinear test-rep × train-rep structure of classical influence functions but dispenses with gradients and Hessian inverses. Instead, it (i) introduces a soft-attention pooling layer to model higher-order, group-level interactions among training points and (ii) jointly learns both the encoder and the pooler on automatically generated pairwise rankings derived from true re-training losses (Datamodel-style supervision). Once trained, AirRep scores influence with a single forward pass, delivering ~80× faster inference and ~50× smaller storage than LoGra while matching or exceeding gradient baselines on fidelity (Linear Datamodeling Score), data‐selection utility, and data-classification accuracy. A single 0.5 B-model–trained checkpoint generalises to 7 B Qwen models and even to LLaMA-1 B without retraining, amortising the one-off training cost after roughly 5 × 10⁵ attributed examples.

**Questions:**

None

**Ethical Concerns:**

["NO or VERY MINOR ethics concerns only"]

**Quality:**

4

**Strengths And Weaknesses:**

## Strengths

- **Principled combination of influence-function theory and Datamodel learning**. AirRep keeps the theoretically grounded bilinear structure of influence functions yet learns the coefficients from ground-truth re-training data, uniting fidelity with efficiency.

- **Attention pooling derived from higher-order influence analysis**. The softmax weighting mirrors coefficients that arise in second- and third-order influence terms, yielding a principled and effective mechanism for group attribution that naïve summation misses.

- **Joint encoder-and-pooler optimisation on ground-truth comparisons**. Auto-generated pairwise rankings align the representation directly with model behaviour, producing task- and model-aware embeddings with no manual labels.

- **Downstream applications/utility**. AirRep  enables data selection (retaining accuracy while pruning the corpus) and data classification (retrieving semantically aligned training points).

- **Thorough empirical validation**.  Experiments cover fidelity, data selection, data classification, ablations, scalability, and cost amortisation across four corpora and five LM sizes, against well-tuned gradient and representation baselines.

- **Strong practical gains and robust generalisation** AirRep achieves ~80× speed and ~50× storage savings over LoGra, with a single checkpoint transferring unchanged to larger Qwen models and to LLaMA-1 B.

## Weaknesses

- **High initial supervision cost.** Building the O(1m) pairwise comparison labels requires training many mini-LMs and forward passes. This can get prohibitively expensive, especially when each run takes a long time to converge.

- **Proxy-model mismatch.** Supervision is gathered on a 0.5 B Qwen model, yet AirRep is evaluated on targets up to 7 B (and on a different architecture). The paper does not quantify how fidelity degrades as the gap between proxy and target widens.

- **Experimental-design limitations.** Baseline gradient methods (TracIn, LESS) are minimally tuned (single checkpoint / LoRA rank), and evaluation stays within instruction-tuning. Stronger baselines and broader domains (code, vision-language, RL) are needed to establish generality.

- **Better iso-FLOP baselines**. To make the cost comparison fully transparent, can maybe include a LoGra (or similar) run whose total compute budget—gradients, Hessian pre-conditioning, plus any pairwise scoring—is matched to AirRep’s cost for generating comparison labels and training the encoder + pooler. This feels like a more apples-to-apples comparison to me.

---

> ### Author Rebuttal · Authors · 2025-07-31
>
> Thank you for your constructive feedback! We appreciate that you find our method principled and effective, our experiments thorough, and the performance is strong. We would like to respond to your comments as follows:
>
> ---
>
> > ### **High initial supervision cost:**
>
> Thank you for raising this concern. In Section 9, we provide a quantitative analysis of AirRep’s initial training cost and demonstrate how it can be amortized during inference. While the upfront cost of training AirRep is indeed higher than that of baseline methods, this is **amortized after approximately 5.6 GPU hours—beyond which AirRep yields an 80× speedup in inference**. Given this substantial gain in downstream efficiency, we believe the one-time offline training cost is a worthwhile investment and does not pose a significant limitation in practical settings where such training time is likely affordable.
>
> ---
>
> > ### **Proxy-model mismatch:**
>
> Thank you for this insightful point. We agree that proxy-target mismatch may lead to fidelity degradation as the model gap widens. To investigate this issue further, we conducted additional evaluations across a range of target models with varying sizes and architectures. The results on FLAN are summarized below:
>
> Method | Qwen-0.5B | Qwne-7B | Llama-1B | Qwen3-0.6B | Qwen3-0.6B (Thinking) | TinyLlama-1B | GPT-2 |
> | - | - | - | - | - | - | - | - |
> LoGra (Qwen-0.5B) | 19.75 | 7.96 | 16.79 | 11.42 | 17.20 | 18.60 | 9.85 |
> LoGra (Target-LM) | 19.75 | 8.75 | 19.95 | 19.54 | 21.85 | 20.14 | 17.78 |
> AirRep (Qwen-0.5B) | 21.11 | 9.18 | 22.54 | 21.62 | 23.26 | 21.48 | 18.16 |
>
> These results show that AirRep consistently outperforms LoGra, even when supervision is derived from a much smaller proxy model (Qwen-0.5B), suggesting strong generalization across both model scale and architecture. While **degradation is observed as the proxy-target gap increases (e.g., Qwen-0.5B → GPT-2), AirRep maintains robust relative gains compared to LoGra**. Due to limited time for the rebuttal, we will include a more detailed discussion and evaluation on this point in the final version.
>
> ---
>
> > ### **Minimal tuning of baselines:**
>
> Thank you for the thoughtful critique. Regarding baseline tuning, as noted in Appendix E, we carefully optimize both performance and efficiency for LoGra, TracIn and LESS. This included exploring different configurations and applying engineering optimizations on our evaluation data.
>
> - For **TracIn**, we report results using the final checkpoint rather than ensembles, as ensembling across multiple checkpoints linearly increases computation and storage overhead. In our preliminary experiments, we found that ensembling yields modest gains compared to simply increasing the projection dimension, while incurring higher resource costs and latency. We will elaborate on this trade-off more fully in the revised version of the paper.
>
> - For **LESS**, we used the LoRA rank recommended by the original paper. This setting already results in an embedding of dimensionality 8192—over 21× larger than that used by AirRep—making it a resource-intensive baseline.
>
> ---
>
> > ### **Better iso-FLOP baselines:**
>
> Thank you for this valuable suggestion. In **Section 9**, we currently report a full breakdown of the total computational cost—including data generation, training, and inference—for both AirRep and LoGra. Our analysis shows that AirRep becomes cost-effective after approximately 5.6 GPU hours, beyond which it offers significant inference-time speedups (up to 80× faster).
> We note that this comparison represents an upper bound on AirRep’s training cost, as the learned representations can often be reused across different target models or tasks without retraining. This enables cost amortization in multi-task or multi-model scenarios, improving overall efficiency in practical deployments.
>
> Following your suggestion, we will revise the paper to include a clearer iso-FLOP-style analysis by aligning LoGra’s total budget (including gradient computations, Hessian preconditioning, and pairwise scoring) with that of AirRep’s full training pipeline.
>
> ---
>
> **Thank you again for your thoughtful feedback—we hope these clarifications and forthcoming additions will resolve your concerns.**

---

> > ### Comment · Reviewer_VLz2 · 2025-08-04
> >
> > Thank you for the rebuttal! This resolves my main concerns.

---

> > > ### Author Response · Authors · 2025-08-05
> > >
> > > We appreciate your valuable feedback on our paper and are glad to have addressed your concerns.

---

### Note · Authors · 2025-08-16

Dear Reviewers, AC, and SAC,

We are grateful for the reviewers’ recognition that we have addressed their concerns and for their support of our paper. We are committed to further strengthening the work by incorporating their valuable suggestions, including clarifying the DsDM baselines (wX2D), adding proxy-model mismatch analysis (VLz2), including LOO results (32FH, wX2D), clarifying the question discussed with zyXJ, and addressing all other points noted in our rebuttal.

Thank you once again for your thoughtful feedback and consideration!

Authors of Paper 18177

---

### Decision · Program_Chairs · 2025-09-17

**Decision:**

Accept (spotlight)

**Comment:**

This submission proposes a method for training data attribution (TDA) called AirRep, where the core idea is to train a representation encoding model for the task of ranking/comparing training examples according to their influence scores.

Strengths identified by the reviewers are as follows:
1. Learning representations for TDA is an innovative idea and "unites fidelity with efficiency" (quoting Reviewer VLz2).
1. AirRep includes an attention-based pooling module to account for the effects of groups of training examples. Reviewer wX2D appreciated that this contribution was separately evaluated in combination with some of the baseline TDA methods.
1. Large efficiency gains are achieved at inference time.
1. A single AirRep checkpoint trained on a Qwen-0.5B model transfers to other LLMs such as Qwen-7B and LLaMa-1B.
1. Thorough empirical evaluation, covering TDA fidelity (measured by Linear Datamodelling Score, LDS), data selection, data classification, an ablation study, and AirRep's supervision and training cost.
1. The pairwise comparison data used to train AirRep is generated automatically.

Weaknesses:
1. AirRep has a high initial cost for generating training data and performing the training. In their rebuttal, the authors emphasized the results in Section 9, which show for example that AirRep requires $\sim 10^5$ training examples to outperform LoGra and amortizes its training cost after 5.6 GPU hours, after which it can process substantially more TDA queries than LoGra. Still, I think that this indicates a requirement for high-volume TDA applications for AirRep to be advantageous.
1. Several reviewers pointed out that leave-one-out (LOO) evaluation (predicting the effect of leaving out one training example at a time) was missing. The author rebuttal reported on a LOO evaluation for a subset of the experiments. This LOO evaluation could be made more complete.
1. Unclear reasons for including only a subset of the baselines in each experimental section (as well as not including TRAK). The rebuttal promised to clarify and include complete comparisons in the appendix.
1. Limitation of the current work to the modality of language could be acknowledged.

The reviewers reached a consensus that the submission should be accepted, with three reviewers rating the work as a clear Accept based on the many strengths listed above. In my view, the weaknesses are either inherent to AirRep and well-discussed in the paper, or very reasonable to address for the camera-ready version.

I am recommending a spotlight presentation because of strengths 1, 4, 5: The innovativeness of learning representations for TDA, coupled with the thorough evaluation including an analysis of when AirRep's training cost (its main weakness) is amortized, and the transferability between models further mitigating the cost.